

# Development and application of the Round-trip Drifting Sounding System (RDSS)

Xiaozhong Cao[1], Qiyun Guo[2], Haowen Luo[2], Rongkang Yang[2], Peng Zhang[2], Guo Jianping[3], Jincheng Wang[4], Die Xiao[5], Jianping Du[6], Zhongliang Sun[7], Shijun Liu[8], Sijie Chen[9], Anfan Huang[2]

[1] China Meteorological Administration, Beijing, China;

[2] Meteorological Observation Centre of China Meteorological Administration, Beijing, China;

[3] Chinese Academy of Meteorological Sciences, Beijing, China;

[4] Numerical Prediction Center of China Meteorological Administration, Beijing, China;

[5] Hunan Key Laboratory of Near-space Meteo-ballon Materials and Technology, Zhuzhou Research & Design Institute Co, Ltd, Zhuzhou, China;

[6] Beijing Huayun Orient Detection Technology  Co, Ltd.,Beijing, China;

[7] Allystar Technology (Shenzhen) Co.LTD. , Shenzhen, China;

[8] Department of Advanced Technology Training of China Meteorological Administration, Beijing, China;

[9] National Satellite Meteorological Centre of China Meteorological Administration, Beijing, China;

*Corresponding author: Xiaozhong Cao, caoxzh@cma.gov.cn*

**ABSTRACT.** Meteorological sounding primarily refers to the balloon-borne radiosonde, which conducts a ground-to-uppe-rair "ascent phase" sounding. This paper introduces the Round-trip Drifting Sounding System (RDSS), an innovative system characterized by three observation phases—'Ascent-Drift-Descent' (ADD)—in which all three phases of sounding observation are executed through single balloon launch. Several key technologies were successfully developed, including the carrier (zero-pressure dual-mode meteorological balloon), the payload (System-on-Chip (SoC) module for meteorological sounding), air-to-ground data reception and ground-to-air control command transmission. RDSS data processing framework based on 'Internet cloud + Instrument terminal' was established. Data quality control methods and data assimilation techniques of RDSS were also developed. An interactive experiment encompassing observations and forecasting was conducted to evaluate the quality of experimental data at each phase of RDSS. The quality evaluation results indicate that the data quality in the RDSS 'ADD' phases meets the breakthrough targets outlined in WMO CIMO-8. The observation quality of wind and temperature in both the ascent and descent phases meets the ideal targets specified in WMO CIMO-8. A numerical experiment on the impact of RDSS data assimilation on forecasting demonstrated a 2% reduction in precipitation forecast error at 06:00 and 18:00 (UTC), along with an average 1% improvement in precipitation forecast accuracy following the assimilation of RDSS data. Furthermore, a new trajectory prediction method for RDSS, based on CMA-MESO, achieved an average simulated landing-point error of less than 40 km. Notably, the accuracy of first guess positioning and trajectory prediction for Typhoon 'Saola' in 2023 was significantly enhanced through RDSS data assimilation, reducing the average trajectory prediction error by 40%. On January 1, 2024, operational observations using RDSS commenced at four stations in Guangdong, China. Starting in July 2024, an operational experiment at one hundred and twenty-seven stations within the China Meteorological Administration (CMA) was planned, with the goal of achieving full operational capability at all CMA stations by 2026.



## 1. Introduction

The upper-air meteorological sounding system (hereinafter referred to as 'sounding') constitutes a key element within comprehensive meteorological measurement framework. It is responsible for gathering data on various meteorological elements such as temperature, humidity, pressure, wind speed, and wind direction from the ground up to heights of 30 km and beyond (DuBois et al., 2002). This system provides vertically observed meteorological data for weather forecasting, numerical weather prediction, climate projection, scientific research, and the inspection and calibration of ground-based remote sensing equipment (Seidel et al., 2009). Since the 19th century, the balloon-borne radiosonde has served as the primary tool for direct measurements of upper-air meteorological elements below 30 km and is extensively utilized globally (Gallice et al., 2011).

For over a century, these radiosondes have utilized the direct measurement method of 'one balloon launched, one measurement.' where the radiosonde ascends at a certain speed with the balloon expanding in volume due to decreasing air pressure as altitude increases. Upon reaching a specific altitude, the balloon bursts, concluding the measurement process. This methodology confines effective observations to the radiosonde's ascent phase (Thomas et al., 1958). The disposable nature of radiosondes and balloons necessitates significant costs for multiple deployments. Consequently, economic constraints have led to reductions in sounding operations, such as Russia's decrease from twice-daily to once-daily launches in 2015, impacting the forecasting accuracy of numerical prediction models across Northern Hemisphere countries (Tian et al., 2018).

Currently, the temporal resolution of global sounding data remains limited, posing a significant challenge to its capacity in fulfilling the requirements of routine forecasting. A notable concern arises from the scarcity of direct measurement data during periods characterized by frequent severe convective activity, particularly in the hours immediately following noon (Wang et al., 2019). Numerous studies have demonstrated that the frequent acquisition of sounding data can significantly enhance the accuracy of numerical weather forecasting (He et al., 2013). The United States and other countries have successfully assimilated airborne sonde data into their model systems, resulting in a 20%-40% reduction in errors in hurricane trajectory predictions (Stephen et al., 2013; Wang et al., 2015).

Additionally, numerous institutions are actively investigating techniques to obtain multiple radiosonde data points from a solitary balloon launch. Illustrative projects include the multidisciplinary analysis of the African monsoon and the measurement system research and forecasting experiments conducted in the Asia-Pacific region, which utilize balloons to conduct drop soundings as they drift with stable upper-air winds over the ocean (Ratnam et al., 2014).The upper-air balloon system, known as ValBal, developed by the Stanford Space Program in the United States, has accomplished multi-day flights at altitudes ranging from 10 km to 25 km (Sushko et al., 2011). The system maintains altitude by automatically venting gas and dispensing ballast, allowing the latex balloon to gradually rise while slowly drifting. Similarly, the French Space Agency, CNES, has developed an overpressure balloon (SPB) capable of floating in the stratosphere for over three months (Venel Stephanie et al., 2016). Additionally, the Tata Institute of Fundamental Research Balloon Facility (TIFR-BF) in India has also contributed to this field by developing comparable systems (Anand et al., 2021; Vernier et al., 2018; H. Vernier, 2022).The systems delineated in these studies are primarily used for scientific experiments. Nonetheless, due to high costs and limited ascent rates, they are not viable solutions for long-term operational balloon sounding data collection. Furthermore, these balloons' very slow ascent rates (necessary for





managing factors such as heat dissipation, ventilation, and sensor response times) do not meet WMO standards
for balloon ascent speed (WMO No. 8, 2023).
Another conducted an extensive analysis by compiling four years of radiosonde data collected without
parachutes across various seasons and altitudes, highlighting the scientific value of descent data (M. Venkat
Ratnam et al, 2014). On account of the sounding process of ascending and descending at the same station has
restricted its potential in conducting adaptive or targeting observation on typhoon forecast (Tan et al, 2006).
This paper introduce a new sounding technology—the Round-trip Drifting Sounding System (RDSS). This
system can improve the spatial and temporal frequency of soundings and provides an additional vertical profile
as well as maintaining cost-effective, which acquires radiosonde measurement data throughout three phases—
'Ascent-Drift-Descent' (ADD)—all within a single balloon launch.

## 2. Synopsis of the Round-trip Drifting Sounding System (RDSS)

The RDSS was organizational developed by the Meteorological Observation Centre of the China Meteorological
Administration (MOC of CMA) with other relevant domestic units in China (hereinafter referred to as the
'RDSS research team'). It undertakes a three-phase Round-trip upper-air measurement (Fig. 1). In addition to the
ascent phase of the current sounding system, RDSS enables sounding during both the drift and descent phases.
This innovative approach completes three phases of sounding with only a single balloon launch, representing a
significant advancement comparing to the traditional upper-air sounding method that has been utilized for more
than a century.

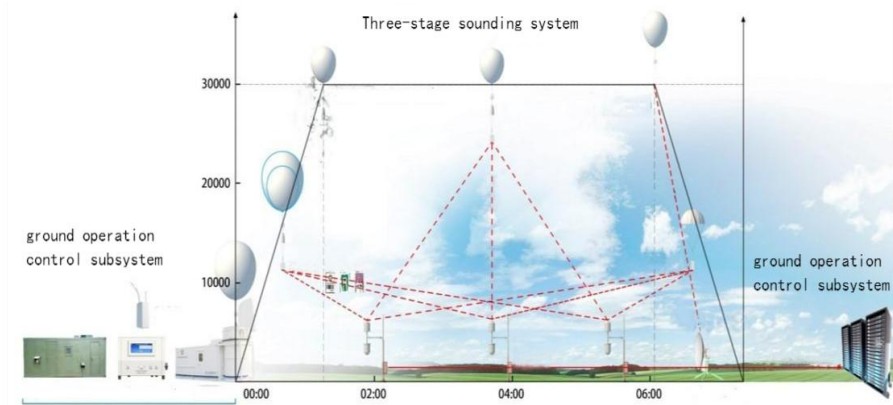


**Figure 1. Operational principle diagram of the Round-trip Drifting Sounding System (RDSS).**
**Table 1. Main equipment and key functions of RDSS.**

| No | | Instruments | Key Function |
|---|---|---|---|
| 1 | | zero-pressure dual-mode meteorological balloon | "outer balloon"as ascent carrier,"inner balloon" as drift carrier |
| 2 | "ADD" subsystem | parachute | parachute as the carrier of the descent phase |
| 3 | | drifting controller | Adaptive control of drift and descent |
| 4 | | radiosonde | The temperature, pressure, humidity, wind measurement meet the demand for long-term stratospheric observation |



| 5 | Ground operation control subsystem | ground station | ground inspection ground check, balloon inflation, launch, and other tasks before the equipment is launched |
|---|---|---|---|
| 6 | | ground data-receiving device | 8 channels receive radiosonde data simultaneously |
| 7 | | control command transmitter | In the weather-sensitive area without a station, the active fusing drifting controller is carried out and the descent measurement is started |
| 8 | | operational management system | Real-time acquisition, transmission, quality control, and timely delivery of control instructions for RDSS data, providing real-time high-quality data to weather analysis and numerical prediction models |

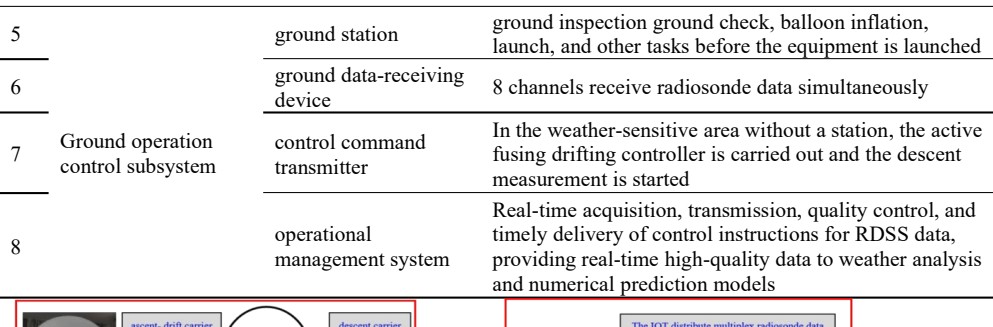

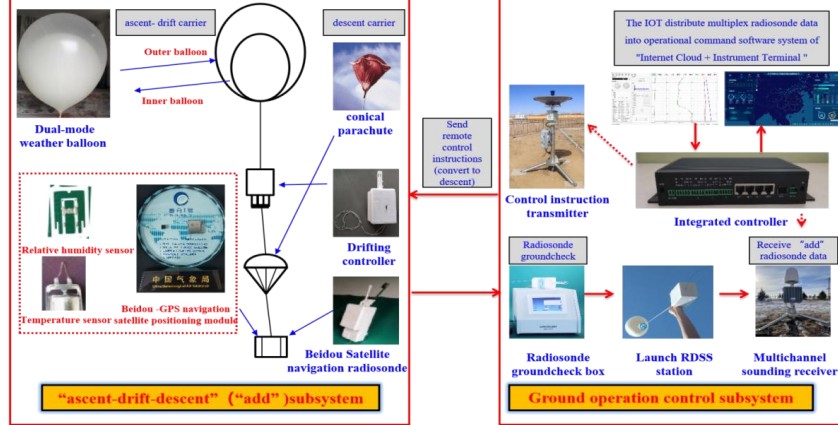

**Figure 2. Schematic representation of the equipment composition for RDSS.**

The RDSS primarily consists of the 'ADD' subsystem and the ground operational control subsystem, as shown in Table 1 and Fig. 2. The 'ADD' subsystem encompasses a zero-pressure dual-mode meteorological balloon with a parachute, a drifting controller, and a radiosonde. The zero-pressure dual-mode meteorological balloon (hereinafter referred to as the 'double-layer balloon') features a design where one balloon embedded within another, both made from a latex material similar to that of conventional meteorological sounding balloons. Upon inflation and launch of the double-layer balloon, the entire 'ADD' system ascends at approximately 400 meters per minute. As it rises, the external air pressure decreases, causing the balloon to expand. At the predetermined altitude (generally between 28km and 30km), the outer balloon bursts due to its expanding volume, marking the conclusion of the ascent phase measurement. Given that the outer balloon bursts within the stratosphere, where vertical air movement is minimal, horizontal movement becomes predominant. The enhanced performance and controlled aeration of the inner balloon enable it to resist bursting. At this juncture, the buoyancy of the inner balloon attains equilibrium with gravity, achieving an approximate vertical stability. Subsequently, the inner balloon, primarily influenced by horizontal air currents, functions as the carrier for the drift phase, which initiates thereafter.

After drifting for a predefined duration, which may vary from a couple of hours to over ten hours, the drifting controller separates the inner balloon from the rest of the RDSS equipment, thereby terminating the drift phase. Subsequent to this separation, the inner balloon persists in ascending until it ultimately ruptures, marking the conclusion of its mission segment. Meanwhile, the remaining components — comprising the parachute and radiosonde—begins to descend. The parachute is promptly deployed, facilitating the radiosonde in collecting



data during the descent phase, while acting as its carrier. This descent persists until the equipment touches down,
thereby completing the final phase of the 'ADD' process. At this point, RDSS has successfully completed the
three-phase 'Ascent-Drift-Descent' measurement cycle.
The ground operation control subsystem of RDSS comprises four main components: the ground station, ground
data-receiving device (downlink communication), control command transmitter (uplink communication), and
the operational management system.
The ground station is similar to existing meteorological radiosonde launch stations, undertaking tasks such as
ground checks, balloon inflation, and launching the balloon with a radiosonde at scheduled intervals. Ground
data-receiving device can also be placed at the launch station. However, its layout and function differ from
conventional-sounding data-receiving equipment. Due to the RDSS drift phase, the horizontal distance between
the descent point of the radiosonde and its launch point can exceed 500km, while conventional radiosonde data
reception has a maximum linear transmission distance of around 200km -300km. Therefore, the traditional
single-station, point-to-point radio communication mode of radiosondes is inadequate for RDSS data reception.
Thus, the ground-to-air communication system has been upgraded from point-to-point to a multiple-to-multiple
model. In areas through which the radiosonde's ADD phases may pass, ground data-receiving devices are
strategically deployed. This configuration enables multiple ground data-receiving devices to concurrently
receive data from a single radiosonde or alternatively, a single receiving device to capture data signals from
several radiosondes simultaneously. Consequently, the ground data-receiving device is designed as a P-band 8-
channel parallel data receiver, capable of receiving data from multiple radiosondes simultaneously. Additionally,
control command transmitters are located at the ground station and other locations. These transmitters send
control instructions from the ground to the drifting controller in the air through uplink communication. This
system allows for the adjustment of the drift phase elevation, termination of the drift phase, and switching to the
descent phase measurement as needed.
The operational management system acts as the brain of the entire RDSS system. Multiple ground data-
receiving devices and control command transmitters are connected to the operational management system via
the Internet. These ground data-receiving devices continuously transmit data to the operational management
system in real-time for processing, display, and storage. Based on the RDSS trajectory and specific weather and
climate conditions, comprehensive decision-making allows the operational management system to transmit
control instructions to control command transmitters which then relay them to the drifting controller in the air to
execute the desired functions.
The 'Internet cloud + Instruments terminal' architecture enables real-time, efficient, and bidirectional
communication across the entire network during the 'ADD' phases. This configuration supports the seamless
real-time acquisition, transmission, and quality control of RDSS data while ensuring rapid data delivery for
weather forecasting. Consequently, it enhances the timeliness and availability of radiosonde data for forecasting
purposes.
**3. Critical scientific problems**
**3.1. Carrier technology**



Compared to using deflation in super-pressure balloons for drifting (Anand et al., 2021; Vernier J. et al., 2018;
H. Vernier, 2022), the double-layer balloon structure of RDSS is simpler, more cost-effective, and suitable for
large-scale deployment in operational upper-air meteorological observations. The ideal 'ADD' process works as
follows: the outer balloon provides lift and explodes after reaching a predefined altitude range. Subsequently,
the inner balloon and its associated equipment achieve vertical equilibrium, enabling a stable drifting state.
Therefore, while the ascent measurement of the balloon sounding is completed, achieving extended drift and
controlled descent remains a technical challenge that RDSS carriers aim to overcome.
**3.1.1 Study on the influence of atmospheric environment on the net lift power of balloons**
A multitude of meteorological factors, encompassing air temperature, air pressure, solar radiation, and other
external environmental conditions, coupled with the gas volume of the outer balloon directly influences its net
lift and burst altitude. Variations in air pressure and temperature within the outer balloon, induced by external
meteorological conditions, interact with the expansion dynamics of the inner balloon. These factors, in
conjunction with the aeration volume of the inner balloon, collectively influence the altitude during the drift
phase and the static equilibrium of the inner balloon. Consequently, controlling the air volume in the double-
layer balloon presents a significant challenge.
The RDSS research team conducted an in-depth theoretical analysis of the ascent and drift processes of the
double-layer balloon, focusing on three areas: upper atmosphere model expansion, the balloon's dynamic
equation, and a thermodynamic model. This study led to the development of a coupling model that accounts for
the effects of atmospheric conditions on the balloon's net lift (from now on referred to as the "coupling model")
(Liu et al., 2022). This model provides a theoretical foundation for determining the net lift force, ascent velocity,
and target burst altitude of the double-layer balloon, enabling precise control of aeration in the double-layer
balloon under varying meteorological conditions.
**Table 2. Aeration test results based on coupling model.**

| Inflatable mode | Effective launch times | Drift number of times | Drift success rate | ≥4h number of times | ≥4h success rate |
|---|---|---|---|---|---|
| Algorithm software | 611 | 479 | 78.40% | 436 | 71.36% |

Table 2 presents data from six stations (Changsha, Wuhan, Anqing, Yichang, Nanchang, and Ganzhou) situated
along the middle and lower reaches of the Yangtze River in 2021. Using the coupling model, we calculated the
aeration capacities of the inner and outer balloons and determined the success rate of the double-layer balloon
launches. The results demonstrate that the coupling model effectively controls the aeration of double-layer
balloon, enabling them to achieve the intended ascent and drift measurements.
**3.1.2 Performance improvement of the double - layer balloon**
The ascent phase of meteorological sounding generally lasts less than one hour. However, during the ADD
process, the inner balloon of the double-layer balloon is exposed to low temperatures, intense ultraviolet
radiation, and high ozone concentrations for several hours or even up to ten hours. To address these challenges,
the RDSS research team conducted formulation tests to enhance the inner balloon resistance to these



environmental factors, with a particular emphasis on natural latex modification, cold resistance, and anti-aging
systems.
Considering that latex hot air aging performance improves air tightness and  balloons are exposed to prolonged
sun exposure and hydrogen loss, the incorporation of a specific latex compound was found to augment
durability. Figures 3a and 3b show that the addition of latex formulas had minimal impact on the latex viscosity
and the balloon appearance while improving its tensile strength and thermal aging resistance. Figures 3c and 3d
demonstrate that butyl oleate exhibits the lowest reduction in low-temperature burst performance after water
extraction and accelerated aging, making it the best-performing cold-resistant agent. Consequently, this agent
was integrated into the formula to bolster cold resistance.
For the anti-aging system, illustrated in Figures 3e and 3f, nano zinc oxide, which can be diluted directly with
water to replace traditional zinc oxide, was used. Additionally, we incorporated antioxidants to enhance the
balloons' resistance against ultraviolet and ozone degradation. These formula improvements extend the inner
balloon lifespan under harsh conditions of low temperature, intense UV radiation, and high ozone levels. This
enhanced durability has been applied to the inner balloon, resulting in an extended service life and a high
success rate during the drift phase (Zhu et al., 2021; Chen et al., 2020).
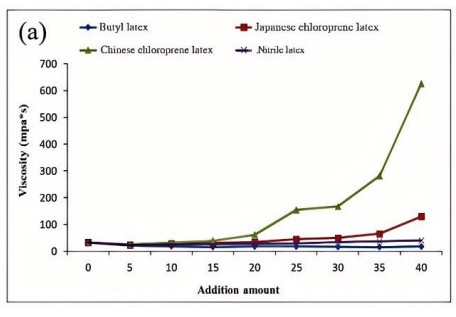
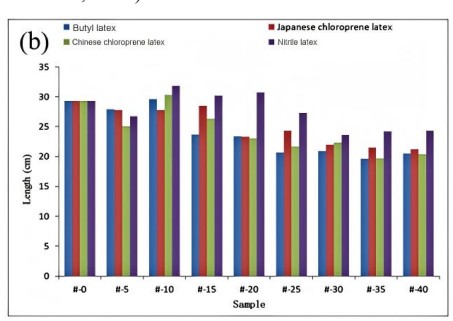

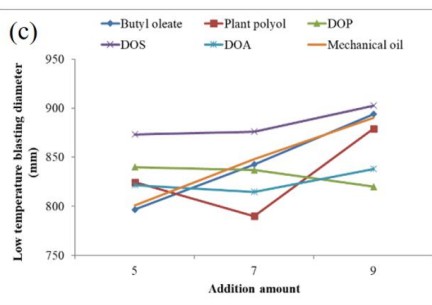
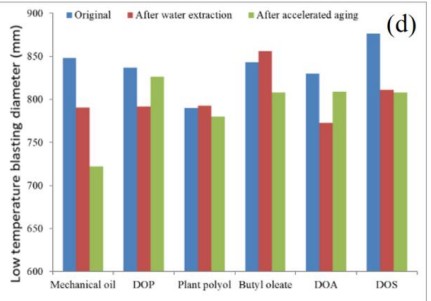

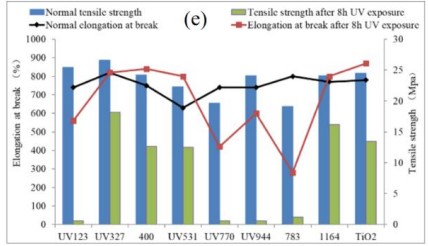
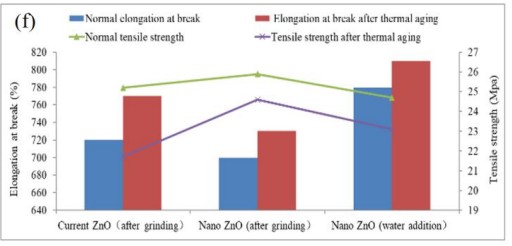





**Figure 3. Modification test of natural latex in the inner sphere: (a) Four types of modified natural latex materials—**
**Butyl latex, Japanese chloroprene latex, Chinese chloroprene latex, and Nitrile latex—were screened and added to**
**natural latex using the homogenization method. (b) Physical properties of the modified natural latex pellet samples**
**were evaluated for conventional and thermal aging tensile testing. (c) Butyl oleate, Plant polyol,DdiOctyl Phthalate**
**(DOP), DiOctyl Sebacate (DOS), DiOctyl Adipate (DOA), and mechanical oil were added to natural latex as cold**
**resistance agents. Cold resistance system test:The blasting diameters of six types of cold-resistant samples were tested**
**using low-temperature blasting instruments at -85°C. (d) Low-temperature blasting diameters were measured for six**
**types of raw, water-pumped, and aged samples with seven parts of cold-resistant agents added. Anti-aging system test:**
**(e) Comparison of tensile properties among nine anti-aging agents—UV327, UV400, UV531, UV1164, and TiO2**
**filler—with 0.2 parts of one-component anti-aging agent after eight hours of conventional and ultraviolet aging. (f)**
**Comparison of tensile properties after ozone aging between Nano ZnO and ZnO.**
**3.2. Payload technology**
**3.2.1 Specialized SoC module of meteorological-sounding**
The RDSS payload is similar to conventional radiosondes and consists of three main components: positioning
systems, PTU (Pressure, Temperature, and Humidity) sensors, and communication modules. It calculates and
outputs satellite positioning information and performs real-time monitoring and transmission of meteorological
data, including temperature, relative humidity, and pressure. Unlike conventional radiosondes, RDSS operates
effectively across all three phases of the ADD process, requiring a minimum operational time of six hours. This
extended operation necessitates a larger-capacity battery for the RDSS radiosonde.
Additionally, the radiosonde's weight affects the balloon's inflation volume, making it essential to reduce the
radiosonde's weight where possible. Therefore, an integrated, lightweight, and low-power radiosonde is crucial,
and the RDSS research team developed a specialized SoC (System on Chip) module for meteorological-
sounding. Based on the ARM Cortex core, this SoC module incorporated RF baseband, worldwide civilian
satellite navigation systems, and sounding data processing in a single chip module that create an efficient low-
power solution. The positioning accuracy of SoC is 0.8 meters horizontally and 1.3 meters vertically,
comparable to the internationally advanced u-blox G7020 module, which achieves 0.7 meters and 1.5 meters
respectively.
**3.2.2 Highly integrated radiosonde**
The GTH3 represents the RDSS's radiosonde, which employs a specialized SoC module of meteorological
sounding and utilizes a multi-layer board design and miniaturized components to reduce the size, weight, and
power consumption. Table 3 demonstrates that the GTH3 radiosonde achieves international standards of
excellence in volume, weight, and transmission power, with advantages in operational duration and
communication rate.
**Table 3. Comparison of parameters among RS41, GTH3 and GTS1 radiosonde.**

| Radiosonde type | Positioning method | Volume (mm³) | Weight (g) | Transmitting power (mW) | Working time (min) | Airspeed (bps) |
|---|---|---|---|---|---|---|
| GTS1 radiosonde (Chinese operational radiosonde) | Radar positioning | 190×90×245 | 400 | >400 | >120 | 1200 |
| Vaisala RS41 | satellite positioning u- | 145×63×46 | 109 | 60 | >240 | 4800 |



| | blox G7020 | | | | | |
|---|---|---|---|---|---|---|
| GTH3(RDSS radiosonde) | Equinox I | 141×66×66 | 90 | 86.3 | >420 | 9600 |

The GTH3 participated in WMO UAII2022(Upper-Air Instrument Intercomparison Campaign organized by the
World Meteorological Organization (WMO) and co-organized by the Deutscher Wetterdienst (DWD) in 2022)
with the results shown in Table 4 (WMO IOM-143). It is suitable for applications in ORUC(Operational and
Research Use in Climatology) , including aeronautic meteorology, near/ultra-short-term forecasting, global
numerical weather prediction, and real-time monitoring.
**Table 4. The evaluation results of GTH3 radiosonde temperature, pressure, relative humidity, wind and geopotential**
**height in WMO Instruments and Observation Methods Report No. 143.(Note: The data are in the form of $\Lambda_{c,L}{}^{\delta_{c,L}}_{\sigma(\delta)} \pm$**
**$\epsilon_{c,L}$ , where $\Lambda_{c,L}$ represents the individual measurement root mean square error, $\epsilon_{c,L}$ denotes the measurement**
**uncertainty, $\delta_{c,L}$ is the measurement error, and $\sigma(\delta)$ indicates the measurement standard deviation. The planetary**
**boundary layer (PBL) ranges from 2 to 7 kilometers; the measurement points above the PBL and below the**
**tropopause 12 kilometers are in the free troposphere (FT); the upper troposphere/lower stratosphere (UTLS) ranges**
**from 7 kilometers to 17 kilometers; the middle and upper stratosphere (MUS) is above 17 kilometers up to the**
**bursting point of the sounding balloon. )**

| GTH3 | Height | Atmospheric temperature [K] | Relative humidity [%RH] | Geopotential height [m] | Pressure [hPa] | Wind (horizontal)direction[°] | Wind (horizontal)speed [ms⁻¹] | Wind (horizontal)vector [ms⁻¹] |
|---|---|---|---|---|---|---|---|---|
| Daytime | PBL | $0.18^{-0.05}_{0.17} \pm 0.03$ | $7.00^{-5.43}_{4.41} \pm 0.74$ | X | X | X | X | X |
| | FT | $0.12^{+0.05}_{0.11} \pm 0.04$ | $8.75^{-3.50}_{8.02} \pm 0.60$ | $5.9^{+2.0}_{5.5} \pm 1.8$ | $0.4^{-0.0}_{0.4} \pm 0.1$ | $3.6^{-0.4}_{3.6} \pm 0.2$ | $0.2^{-0.0}_{0.2} \pm 0.0$ | $0.3^{+0.2}_{0.1} \pm 0.0$ |
| | UTLS | $0.09^{+0.01}_{0.08} \pm 0.03$ | $7.73^{-1.55}_{7.58} \pm 0.40$ | $13.2^{+10.0}_{8.6} \pm 3.8$ | $0.4^{-0.3}_{0.2} \pm 0.1$ | $2.5^{-0.2}_{2.5} \pm 0.3$ | $0.2^{-0.0}_{0.2} \pm 0.0$ | $0.3^{+0.2}_{0.2} \pm 0.0$ |
| | MUS | $0.27^{-0.22}_{0.16} \pm 0.10$ | $1.69^{+1.48}_{0.82} \pm 0.46$ | $29.5^{+23.4}_{17.9} \pm 4.2$ | $0.3^{-0.2}_{0.1} \pm 0.0$ | $6.1^{-0.4}_{6.1} \pm 0.2$ | $1.3^{-0.0}_{1.3} \pm 0.0$ | $1.5^{+0.3}_{1.5} \pm 0.0$ |
| Nighttime | PBL | $0.38^{-0.18}_{0.34} \pm 0.05$ | $4.72^{+0.74}_{4.66} \pm 0.15$ | X | X | X | X | X |
| | FT | $0.15^{+0.02}_{0.15} \pm 0.02$ | $6.41^{+2.16}_{6.03} \pm 0.11$ | $5.8^{+0.4}_{5.8} \pm 0.4$ | $0.5^{+0.1}_{0.5} \pm 0.2$ | $2.6^{-0.2}_{2.6} \pm 0.2$ | $0.2^{-0.0}_{0.2} \pm 0.0$ | $0.2^{+0.2}_{0.1} \pm 0.0$ |
| | UTLS | $0.12^{+0.06}_{0.10} \pm 0.05$ | $6.82^{+3.70}_{5.74} \pm 0.26$ | $11.5^{+7.7}_{8.6} \pm 3.4$ | $0.3^{-0.1}_{0.2} \pm 0.1$ | $2.4^{-0.1}_{2.4} \pm 0.1$ | $0.2^{+0.0}_{0.2} \pm 0.0$ | $0.2^{+0.2}_{0.1} \pm 0.0$ |
| | MUS | $0.10^{-0.03}_{0.10} \pm 0.02$ | $1.71^{+1.54}_{0.74} \pm 0.28$ | $26.7^{+20.7}_{16.8} \pm 4.2$ | $0.1^{-0.1}_{0.1} \pm 0.0$ | $4.5^{-0.6}_{4.4} \pm 0.2$ | $0.2^{-0.0}_{0.2} \pm 0.0$ | $0.4^{+0.3}_{0.3} \pm 0.0$ |

**3.2.3 Drifting controller**
The drifting controller can be considered part of the RDSS payload. It connects to an inner balloon above and a
parachute and radiosonde below. The controller fuses a wire with an instant high electric current, triggering the
mechanical device to disconnect the parachute and the radiosonde. The controller serves two main functions
during the drift phase:
1.Reduce the weight of the sounding equipment by releasing a pre-carried counterweight, in order to adjust the
gravity or altitude of the inner balloon and its load during the drift phase.



2.Separate the inner balloon from the other sounding equipment (parachute and radiosonde). Moreover, the
drifting controller can also initiate the fuse based on predetermined control rules, such as altitude limits (≤18
km), specified time, the latitude and longitude of a designated area, or upon receiving commands from the
ground. This action can be taken before the drifting balloon is about to enter the specified area or approach the
maximum drift height. As a result, it will effectively end the drift phase, separating the parachute and
radiosonde from the balloon.

### 3.3. Receiving radiosonde data and sending control instructions technology

The RDSS ground data-receiving device  utilizes a high-gain, low-power, ultra-compact omnidirectional
antenna, along with super heterodyne architecture and multiple communication protocol algorithms, including
time division multiple access, frequency division multiple access, and code division multiple access (Gong et al.,
2021). It supports various frequency modulation modes and achieves a receiving sensitivity of better than -120
dBm (at 2400 bps), effectively addressing self-interference issues in multi-channel radiosonde data reception.
Additionally, the system incorporates narrow-band wireless communication technology to improve low-
elevation reception when the radiosonde drops below the receiving antenna's height, facilitating broad-area
coverage with a visual range radius for upper-air coverage of at least 200 km. The receiver can adapt to diverse
application scenarios, such as fixed stations, vehicles, and ships. With an average data reception rate of 99.7%,
the ground receiver at the Anqing station has demonstrated an impressive maximum reception distance of up to
487 km.
Unlike the one-way (downlink) communication mode used in conventional sounding systems, the RDSS control
command transmitter can send ground instructions to the drifting controller, with a linear communication range
extending beyond 300 km. This capability allows for precise control over the drifting controller to execute
actions such as releasing counterweights or separating the balloon from the parachute and radiosonde, enabling
the radiosonde to conduct drift phase measurements within the target area (Liu et al., 2021). During field tests,
over ten balloon discharge control commands were successfully transmitted, with the farthest reaching 403 km.

### 3.4. "ADD" measurement technology

The details of the ADD measurement method are outlined by Cao et al. (2019). The ascent phase measurement
technique adheres to the guidelines outlined in the Guide to Operational Upper-Air Meteorological Observation
(CMA, 2010). The primary research focus of the RDSS is on the measurement techniques for the drift and
descent phases.

### 3.4.1 Temperature measurement method in the flat drift phase

During the drift phase of RDSS, the inner sphere of the double-layer balloon moves with the horizontal airflow
in the stratosphere. The radiosonde's vertical movement, with the surrounding atmosphere, is minimal and can
be approximately considered as drifting with the horizontal wind. The effect of radiation on the temperature
sensor during this phase is significantly greater than during the ascent and descent phases, leading to
considerable measurement errors that are challenging to correct using general quality control algorithms. For
instance, Vaisala radiosonde software often flags such data as invalid.



Given the unique conditions of stratospheric air temperature measurement, the RDSS research team employs a
multi-physical field, fluid-structure coupled computational fluid dynamics (CFD) approach to model the
behavior of the temperature sensor in high-altitude, low-wind-speed environments. This model calculates the
flow and the temperature field, accounting for radiation effects based on sun elevation, ventilation, sensor size,
and surface reflectivity. To ensure broad applicability, neural networks, and other mathematical methods are
used to fit the extensive simulation data, yielding practical error-correction equations.
Considering that there may be discrepancies between CFD simulations and real environmental conditions, the
RDSS research team uses instruments such as low-pressure wind tunnels and solar simulators to create an
experimental platform. This setup simulates ventilation, air density, and solar radiation conditions during the
drift phase, allowing for the measurement of temperature errors due to solar radiation. These measurements
verify and refine the simulation-based error correction equations (Yang, R.K., 2014).

### 3.4.2 Vertical wind data extracted by parachute landing

Currently, due to the pendulum effect, vertical wind measurements cannot be performed in balloon-borne
soundings. The parachute-drop wind measurement model established by the WMO and NCAR does not address
vertical wind measurement directly,  instead assuming a zero vertical wind speed. This model assumes that the
parachute-drop system is influenced only by gravity and vertical resistance, omitting other factors like buoyancy,
additional forces, and parachute rotation during descent. This limitation prevents the analysis of vertical wind
and also affects the accuracy of horizontal wind field calculations (Li, 2010).
Therefore, the RDSS research team developed a more comprehensive vertical wind measurement model by
considering all relevant forces acting on the parachute descent system. Comparative tests led to the selection of
conical parachutes to minimize swing effects on measurements, with the parachute area tailored to match the
radiosonde weight. As a result, RDSS achieves a stable descent speed of approximately 6 m/s ± 1 m/s, a swing
angle  below 5°, and a vertical wind measurement uncertainty of less than 1 m/s. These findings demonstrate
that the model is effective for calculating vertical wind (Guo, 2018).

### 4. Field experiments and data quality verification

### 4.1. Field experiment in the middle and lower reaches of the Yangtze River Region

From 2019 to 2021, RDSS conducted field tests and application research across a wide area in the middle and
lower reaches of the Yangtze River in China. The research focused on measurement data processing methods,
quality control algorithms, and application technologies across diverse scenarios. The RDSS network spans six
stations: Anqing, Wuhan, Yichang, Nanchang, Changsha, and Ganzhou, located in Anhui, Hubei, Jiangxi, and
Hunan provinces. A total of 14 ground radiosonde data receivers were strategically positioned around these six
stations, spaced approximately 150 km apart. The test covered an area of 600,000 square kilometers, as shown
in Fig. 4, and spanned a duration of 13 months ,inclusive of seven consecutive months of testing from March to
September 2021.



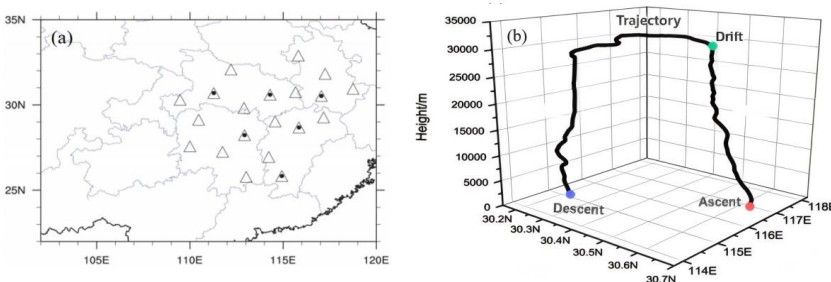


**Figure 4 (a) The network distribution of RDSS and (b) an example of measurement: the trajectory for RDSS at the**
**Anqing station at 12:00 UTC on 11 June 2018. The black dots represent RDSS stations, while the triangles represent**
**ground radiosonde data receiving instruments.**

During the 13-month experimental period, 3,177 RDSS launches were conducted, with 3,012 classified as
effective launches, of which 2,369 achieved successful drifting. Among these, 2,136 launches resulted in
drifting for more than 4 hours. The overall drifting success rate was 78.65%, with a 4-hour drifting success rate
of 70.91% (Fig. 5). During the experiment period, the balloon ascent reached a maximum altitude of 38.3 km
and a minimum of 18.1 km, with an average ascent height of 26.8 km. The recorded drifting distances varied
from a minimum of 24.5 km to a maximum of 684.3 km, yielding an average drifting distance of 219 km. The
longest drifting duration was 616 minutes, and the shortest was 80 minutes, with an average drifting time of
282.5 minutes.

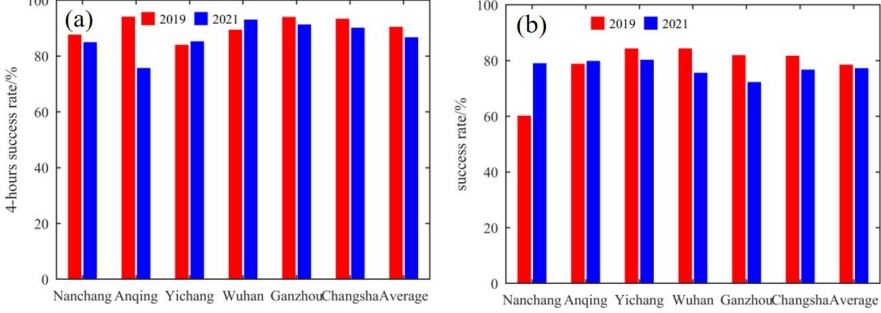

344

**Figure 5. The sounding-forecasting interactive network experiment (2019-2021): (a) Drifting success rate; (b) 4-hour**
**drifting success rate.**

Due to the limited availability of high frequency, continuous measurement data for the stratospheric atmosphere,
experiments were conducted in the middle and lower reaches of the Yangtze River to obtain direct measurement
data with high spatial and temporal density(Zhang et al. 2021). These data provide valuable support for studying
the interaction mechanisms between the troposphere and the stratosphere (Fig. 6).



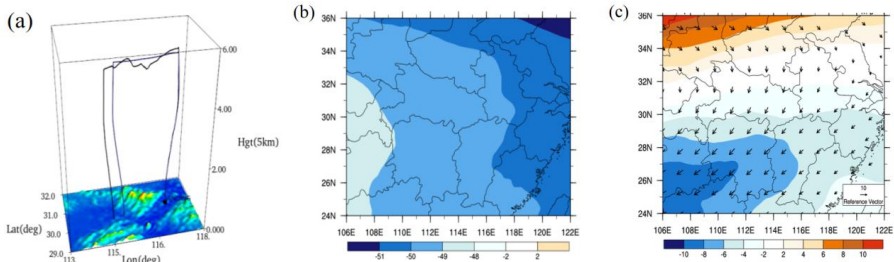


**Figure 6. (a) RDSS three-dimensional trajectory diagram; (b) Temperature field at 200 hPa in the middle and lower**
**reaches of the Yangtze River; (c) Wind field at 200 hPa in the middle and lower reaches of the Yangtze River.**
**4.2 Data quality assessment**
Aiming at the characteristics of high-resolution RDSS data for quantitative application, the RDSS research team
carefully evaluated RDSS data using 31 data quality control methods based on the guidelines for operational
upper-air meteorological observation (CMA, 2010) (Wang et al. 2020). For the data quality of the ascent phase
of the ADD, refer to the results from the UAII2022 of GTH3 (WMO IOM-143). Additionally, the fifth
generation of ECMWF (ERA5) was used to evaluate the QC data of RDSS (Minola et al. 2020; Roja et al. 2011),
and Table 5 shows the results. Compared with the data in the ascent phase and the descent phase, the results for
temperature, u-wind, v-wind, and the standard deviation of geopotential height show good consistency, relative
humidity shows weaker uniformity. The conclusion is basically consistent with the results of Table 4 in the
section 3.2.2.
**Table 5. Comparative analysis of after-quality control of RDSS "ADD" radiosonde data and EAR5.**

| phase | Vertical hierarchy | The standard deviation of RDSS "ADD" radiosonde data and ERA5 was compared and analyzed | | | |
|-------|--------------------|------------|------------|------------|------------|
|       |                    | U (m/s) | V (m/s) | T(K) | RH (%) |
| Ascent | below troposphere top | 1.28 | 1.44 | 0.92 | 8.99 |
|        | above troposphere top | 1.44 | 1.72 | 1.47 | — |
| Drift | above troposphere top | 3.32 | 3.22 | 3.09 | — |
| Descent | below troposphere top | 1.51 | 1.61 | 0.80 | 8.65 |
|        | Above troposphere top | 1.29 | 1.73 | 1.20 | — |

**5. Application of RDSS in numerical forecasting techniques**
The drifting trajectories of RDSS radiosondes were all from west to east (Fig. 9), which aligned with the
movement and development direction of severe convection. Therefore, the descending RDSS radiosondes can
track the occurrence of the entire convective system and the changes in the ambient field during the



development of the convective system in real-time. This capability provides effective monitoring and significant
insights into the environmental conditions favorable for the occurrence and development of convection.

**5.1 The applications in weather analysis**

Through long-term testing, the 'ADD' system can better capture key information in weather system monitoring.
From July 8 to 9, 2021, a strong convective weather event with a long duration and a large impact area occurred
in the middle and lower reaches of the Yangtze River in China. Convection developed and moved to northern
Jiangxi, northern Zhejiang, southern Anhui, and southern Jiangsu overnight on August 8 (Fig. 7).
At 19:15 on the night of the 8th, Wuhan was located on the west side of the main body of the convection, and
the GTS1 radiosonde was launched from the Wuhan station. Subsequently, at 20:00, the RDSS radiosonde
(GTH3) was also deployed from the same location. The trend of the radiosonde curve in the ascent phase (Fig.
8b) mirrored the trend observed for the GTS1 radiosonde (Fig. 8a). It was evident from the radiosondes in the
ascent phase that the CAPE values of the GTH3 and GTS1 radiosondes differed with the development of
convection. The CAPE value of the GTH3 at 20:00 was 1437.7 J/kg, exceeding the 926.6 J/kg detected by
GTS1 at 19:15. Influenced by an upper-level westerly jet, the RDSS sounding balloon drifted eastward toward
the Huanggang, Hubei Province.
At 21:30, Huanggang exhibited higher humidity and more intense convective activity compared to Wuhan. At
this time, RDSS conducted descent phase measurements (Fig. 8c). As shown in the layer curve, the CAPE
value at 21:30 decreased significantly compared with that at 20:00, dropping to 559.7 J/kg, which was lower
than the energy recorded at the Wuhan station. Given Huanggang's proximity to the main body of convection,
the reduction in energy suggests a weakening of upward motion, leading to the accumulation of effective
potential energy and further suppression of convective development. Notably, the GTS1 radiosonde, launched
from Wuhan, ceased data collection after the ascent phase, thus missing this crucial change.
The drifting trajectories of RDSS radiosondes in the middle and lower reaches of the Yangtze River in China
from July to August 2021 (Fig. 9) uniformly followed a west-to-east path, aligning with the region's strong
convective direction.
The descent phase of RDSS demonstrates robust monitoring capabilities and holds significant implications, as it
can timely captures environmental conditions favorable for convection onset and development, including wind
patterns, effective potential energy, and humidity at downstream locations.. These findings enable researchers to
analyze changes in the upper-air field and the occurrence of catastrophic weather convective systems (Zhang et
al. 2021).



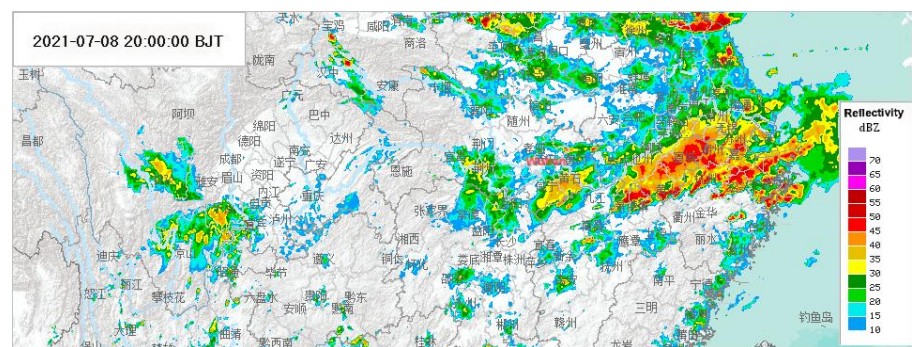


**Figure 7. Combined Reflectivity Factor of Radar Mosaic in the Yangtze River Basin, China, at 20:00 on July 8, 2021.**

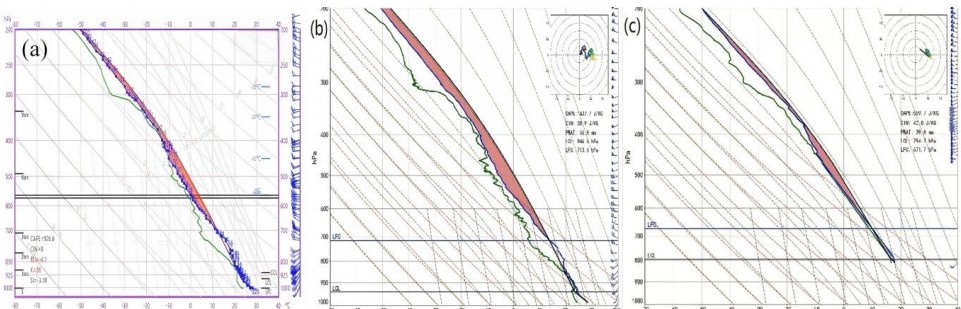


**Figure 8. Comparison of RDSS GTH3 and GTS1 radiosonde T-lnP at the Wuhan station: (a) Wuhan Balloon**
**Sounding at 19:15; (b) Wuhan RDSS ascent phase at 20:00; (c) Wuhan RDSS descent phase at 21:30.**

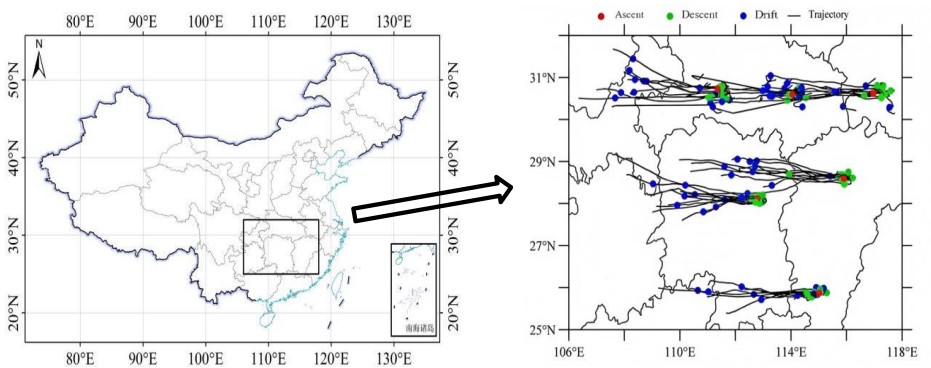


**Figure 9. The RDSS drifting trajectory from July to August 2021.**
**5.2 Applications in numerical weather prediction (NWP)**
Compared with operational radiosondes, the RDSS adds descent phase measurement data for numerical weather
prediction (NWP), achieving a similar role to intensive sounding and providing more continuous, direct
stratospheric measurement data. The Numerical Department of the China Meteorological Administration
developed the key technology for RDSS assimilation in the CMA-MESO 3DVar and CMA-GFS 4DVar systems
(Gong et al. 2019). With the four-dimensional ensemble forecast error is introduced into the CMA global data





assimilation system, and the H-4DEnVar assimilation scheme is developed. The batch cycling forecast
experiments and typhoon forecast experiments are conducted and compared with the 4DVar scheme (Wang F et
al, 2024). Specifically, this includes observation operators that consider drift positions and vertical sparring
methods, such as selecting the nearest radiosonde data from the model layer for assimilation (Guo et al. 2018).
We employed the CMA-MESO V5.1 to conduct a measurement data assimilation in the RDSS descent phase
across six test stations in the middle and lower reaches of the Yangtze River from July 1 to July 31, 2021. We
set up the control test (CTL)  as in the CMA-MESO service system, and the observed data included traditional
sounding data, ground reports, aircraft reports, cloud-guided wind, radar radial wind, GNSS occultation
refractive index, and ground-based GNSS retrieval of the atmospheric whole-layer precipitable water. RDSS
data assimilation was added to the control CTL in the Down test.
The impact of the RDSS descent phase measurement data on the precipitation forecast at CMA-MESO at 03, 06,
09, 15, 18, and 21 UTC (termed the warm start times) was evaluated. Compared to the TS (Threat Score), the
ETS (Equitable Threat Score) imposes stricter penalty for false alarms and missed reports, making the scoring
more equitable. The results of the one-month batch test indicate that assimilating RDSS descent phase data
improves precipitation forecasting skills, especially for heavy precipitation above a certain magnitude. Figures
10a and 10b illustrate the improvement rates in accumulated precipitation forecasting skills for the 0-12 hour
and 12-24 hour periods from the warm start time. Positive values indicate that the precipitation forecasting skills
of the Down test are improved compared with those of the CTL test, while negative values indicate a decrease in
forecasting skills for the Down test.
The ETS scores for precipitation forecasts in the 0-12 hour range at thresholds of 0.1 mm, 1 mm, and 50 mm
increased slightly, averaging about 0.04% (Figure 10a). Due to the timeliness required for forecasting, the 12-24
hour precipitation forecast is of particular interest to forecasters. As illustrated in Figure 10b, the Down test
demonstrated enhanced ETS scores for precipitation forecasts across all levels within the 12-24 hour range, with
an average increase of 0.7% at the 50 mm threshold and a notable 2.2% improvement specifically at this
level  (Zhuang et al. 2022).
In addition, we utilized CMA-MESO V5.1 to conduct Observing System Simulation Experiments (OSSE) under
the RDSS network nationwide. The results indicate that once the RDSS network observation is implemented,
the national precipitation forecast skills of the CMA-MESO fast cycle assimilation forecast system at warm
startup time can improve by 2%-5%. The potential operational applications of RDSS high-resolution data were
quantitatively evaluated using a numerical model (Wang et al. 2023). After the application of RDSS data in
CMA-GFS 4DVar assimilation, the temperature analysis error at 06:00 and 18:00 was reduced by more than 2%
and the average prediction skill of the CMA-MESO accumulated precipitation results for the 12-36 hour period
improved by 1%(Wang, J. C. et al. 2024).



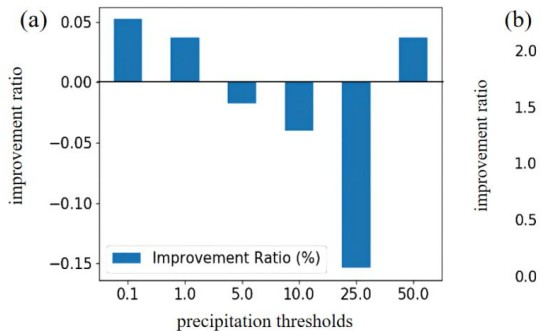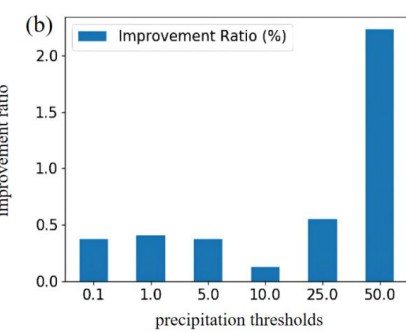

**Figure 10. Improvement rates of cumulative precipitation predictions for 0-12 hours (a) and 12-24 hours (b) in the Down test compared to the control test.**

## 5.3 Applications in targeted observations

Targeted observations have always been a frontier field in atmospheric science research. They represent an important method to address the shortcomings of operational observation systems in monitoring extreme weather events. Furthermore, they significantly enhance the initial field quality and forecast accuracy of numerical models, which is crucial for predicting extreme weather disasters (Majumdar & Sharanya, 2016).

With its capacity for "ADD" measurement, the RDSS has the potential to conduct targeted observations in uninhabited areas, rarely observed regions, and during specific extreme weather events. However, since the RDSS lacks a power system, accurate trajectory prediction is essential for utilizing the descent phase for vertical measurements in these locations. This requires careful consideration of appropriate drift height, launch time, and launch location, allowing the RDSS sounding to be carried to the target observation area by the ambient wind field.

In this context, a trajectory prediction and selection method based on high-resolution numerical weather prediction technology has been proposed for RDSS (Wang et al., 2021).

### 5.3.1 Trajectory prediction method and software system

This paper addresses the issues of low temporal resolution and prediction accuracy associated with the linear extrapolation method used in balloon trajectory prediction. The balloon trajectory equation is directly embedded into a high-resolution numerical weather model system that utilizes a model atmospheric environment with high temporal resolution (1-10 seconds) and high spatial resolution (1-3 km). This approach enables precise simulation of vertical velocity during the RDSS descent phase (Fig. 11a), significantly enhancing the accuracy of RDSS trajectory prediction and the simulation of descent velocity. The average prediction error for a 6-hour trajectory is less than 40 km (Fig. 11b).

Based on this improvement, the RDSS trajectory prediction and simulation system henceforth referred to as the trajectory prediction system, has been established using the CMA-MESO numerical weather model, allowing for multi-station trajectory simulations (Fig. 12a and 12b) (Wang et al., 2020).



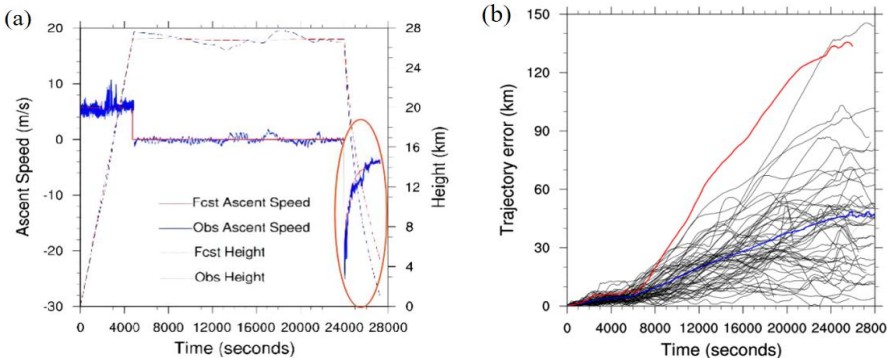

472

**Figure 11. (a) Comparison of simulated (red line) and observed (blue line) vertical speeds of RDSS radiosonde data during the descent phase at the Anqing station at 11:17 on June 20, 2018; (b) Deviations of 63 pairs of simulated RDSS trajectories versus observed trajectories (black line), with the average deviation indicated by the blue line and the largest forecast deviation shown by the red line.**

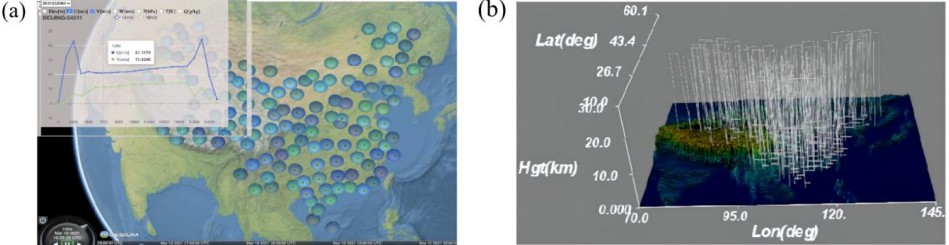

**Figure 12. (a) RDSS trajectory and simulation data display software system; (b) RDSS trajectory forecast chart for China.**
**5.3.2 Trajectory selection method based on the collection idea**
To observe the RDSS in the target observation area, we proposed a method of elevation selection based on
ensemble forecasting, considering the characteristics of the atmospheric wind field as it varies with altitude. The
main idea of this method is to predict the trajectories of all RDSS stations at different drifting heights and select
the heights closest to the target observation area. The details are as follows:
1.    Identify the positions and launch time: Given the positions $S_m(x_m, y_m, z_m)$ of M RDSS launch stations and
the launch time $t_r$    .
2.    Select safe drift heights: Choose N safe drift heights $h_1, h_2... .h_n$, that comply with civil aviation safety
regulations as flat drift heights.
3.    Trajectory prediction: Utilize the trajectory prediction system to predict the RDSS trajectories under the
above conditions within 12 hours, resulting in $T_{nm}(x,y,z,t)$ for N trajectories at each of the M launch
stations.
4.    Calculate closest trajectory: From the M×N trajectories obtained in step 3, calculate the trajectory closest
to the target observation point. When the distance is less than the predetermined standard distance $L_C$ that
can meet the requirements, the releasing RDSS station and the drifting height $H_s$ are selected, and the time
nearest to the target area is taken as the descent time $t_s$. If no suitable drift height meets the conditions, the
trajectory selection fails.



5.    Implement target observation: Input the information regarding the RDSS launch station, drift height, and

498         descent time determined in step 4 into the RDSS operation command system to execute the target

499         observation.

### 500 5.3.3 Targeted observations experiment of Typhoon

The RDSS research team proposed a 'full chain' implementation for target observation using RDSS (Figure 13).
This implementation plan, designed to provide technical support for RDSS applications in disaster weather
monitoring, forecasting, and mechanism research, encompasses three primary stages.
Initially, the requirements for target observation are established. These requirements fall into two categories:
one focusing on specific disaster weather events and the other on sensitive areas to improve future numerical
prediction skills. The target observation area is then determined based on the type of demand. For the first
category, the specific location of anticipated disaster weather is identified through numerical prediction results.
For the second category, the target observation location is determined using CMA-GFS singular vector
technology.
Subsequently, the trajectory selection system is invoked to ascertain the RDSS discharge station, drift height,
discharge time, and other relevant information. This information is then transmitted to the operation command
system of RDSS to guide the stations in implementing RDSS measurements. Ultimately, the RDSS target
observation data is distributed to users for application and evaluation.

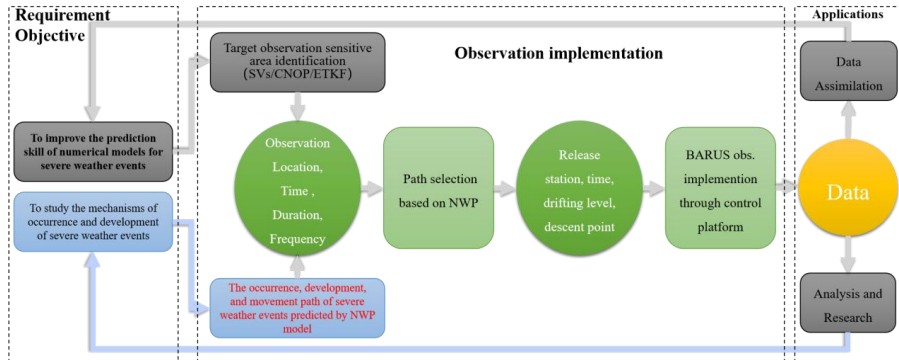


**Figure 13. Technical route for targeted observations of typhoons and other severe weather using RDSS and CMA-**
**MESO models.**
According to the implementation plan for RDSS target observation, we made a preliminary attempt to conduct a
target observation experiment on Typhoon 2309 'SAOLA' (Lau DS et al., 2024) formed at 00 UTC on August
28, 2023. By 00 UTC on September 1, it was expected that 'SAOLA' would land near Guangdong on September
2. Therefore, the demand for vertical profile data of the internal interface of Typhoon 'SAOLA' became
imperative.
Using the typhoon trajectory predicted by CMA-GFS, we pinpointed the typhoon's position for 12:00 PM on
September 2 post-landfall. The RDSS trajectory selection system was then engaged to ascertain the launch
station and drift height that could reach or come closest to the typhoon area, ranging from the minimum
navigation safety height of 21 km to 29 km. We set ten different drift levels at 1 km intervals, with trajectory



predictions and simulations conducted from four stations in Guangdong. Yangjiang station in Guangdong was
ultimately chosen for the launch, scheduled for 06:00 on September 2, 2023, with a drift level of 25 km.
We calculated the required air capacity for the double-layer balloon, and the RDSS 'ADD' subsystem was
prepared to be deployed by station personnel. When the radiosonde reached the core area of 'SAOLA,' the
radiosonde dispatched commands to the 'SAOLA' controller via control command transmission equipment,
successfully observing the descent section 80 km from the center of Typhoon 'SAOLA.' The obtained RDSS
data was subsequently assimilated into the CMA_MESO 3DVar system. Late test results indicated that after
assimilating the data from the RDSS descent section, the forecast error for the typhoon trajectory reported since
06:00 on September 2, 2023, was significantly reduced. Specifically, the typhoon trajectory error at 02:18 was
reduced from 62.7 km to 35 km in the control test ,marking an improvement of 44.18%. Additionally,
precipitation forecasting techniques exhibited significant improvements: from 0.25 to 0.30 in the 10 mm scale,
from 0.30 to 0.55 in the 25 mm scale, and from 0.45 to 0.70 in the 50 mm scale. These results effectively
demonstrate the potential of RDSS in target observation and numerical assimilation applications (Figure 14).
It is worth noting that the initial use of RDSS for target observation served as a foundational attempt, paving the
way for future RDSS operations and maximizing its utility.


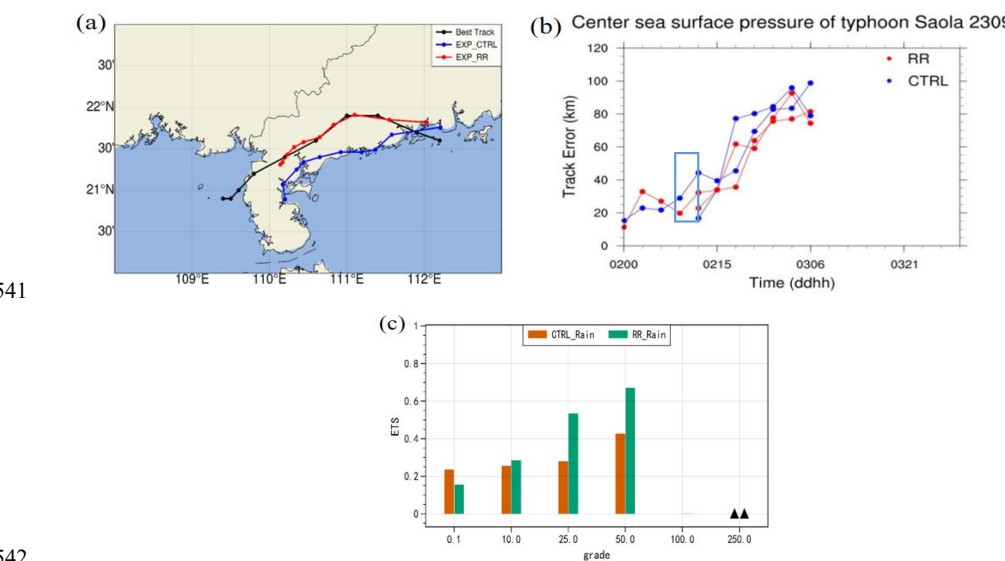

**Figure 14. 2023-09-02 UTC 00-12h: (a) Control test (blue), RDSS data assimilation impact test (red), and optimal**
**trajectory of Typhoon 'Saola' (black); (b) Comparison of sea level pressure at the central point of Typhoon 'Saola'**
**between the control test (blue) and the RDSS data assimilation impact test (red); (c) ETS scores for 0-24 hours of**
**precipitation forecast from control trials (orange) and RDSS data assimilation impact trials (green).**
**6. Summary**
Distinct from traditional operational balloon soundings which focus solely on the ascent phase, RDSS achieved
a three-phase sounding at once by effectively incorporating stratospheric horizontal drift and descent soundings.
This innovation system created a new model for ADD three-phase sounding. Compared to intensive soundings,
RDSS significantly reduces the costs while including stratospheric drifting sounding.



RDSS represents a next-generation approach to acquiring upper-air data, surpassing the century-old
conventional method. We developed a multi-station real-time reception system utilizing 'Internet cloud +
Instruments terminal' technology. Additionally, uplink commands can be sent from the ground to facilitate
descent measurements in designated areas and targeted observations in weather-sensitive regions.
RDSS is essentially a mature system. Following over five years of extensive research and numerous field tests,
the instruments, software, and operational guidelines of the system have achieved a refined level of maturity.
Starting January 1, 2024, RDSS will undergo operational experiments at four stations in Guangdong, China.
Since July 2024, a planned operational trial at 127 CMA stations aims to achieve full operational capability
across all CMA-stations by 2026. RDSS is a situational profiling technique that offers cost-effective upper-air
measurements, making it suitable for widespread application in operational soundings. However, challenges
remain, such as improving the drift success rate, enhancing relevant technologies, and fully leveraging the
potential of continuous measurement data during the drift phase. With the constant development of RDSS and
the continuous deepening of the measurement data, RDSS will become a development direction for future
operational applications and scientific research.

Data availability. Requests for data that support the findings of this study can be sent to luohw_1@163.com.
Xiaozhong Cao1, Qiyun Guo2, Haowen Luo2, Rongkang Yang2, Peng Zhang2, Guo Jianping3, Jincheng
Wang4, Die Xiao5, Jianping Du6, Zhongliang Sun7, Shijun Liu8, Sijie Chen9, Anfan Huang2

Author contributions. XC, QG and HL designed the experiments and wrote the paper; JW, DX, JD, ZS, SL
carried out the experiments; RY, JW analyzed the experimental results. PZ, GJ, SC, AH revised the paper and
participated in the discussion.

Competing interests. The contact author has declared that none of the authors has any competing interests.

Disclaimer. Publisher's note: Copernicus Publications remains neutral with regard to jurisdictional claims made
in the text, published maps, institutional affiliations, or any other geographical representation in this paper.
While Copernicus Publications makes every effort to include appropriate place names, the final responsibility
lies with the authors.

Financial support. This research has been supported by the National Natural Science Foundation of China (grant
no. U2442214) and the National Key Research and Development Program (grant no. 2018YFC1506200) and .




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
