# Peer review of "Development and application of the Round-trip Drifting"

_EGUsphere, 2025_

## Community Comment (CC2)

**Reply TO 'Comment on egusphere-2025-2012', Anonymous Referee #1, 11 Jul 2025**

**Response to Referee #1's comments**

*Referee #1: egusphere-2025-2012*

*Synopsis: The manuscript describes a novel upper air balloon observing system, consisting of a double balloon setup that allows for measurements during a long drift phase in the stratosphere and during the descent  phase. The start of the drift phase and of the descent phase can be triggered remotely and steering the height is possible as well. The manuscript also describes the potential for using the system for targeting observations and how assimilation of the data improves analysis and forecasts.*

*1.Overall comment: It is certainly desirable to make better use of weather balloons than is currently the case with many conventional radiosondes where only data collected during the ascent phase are used. The longer residence of the balloons in the lower stratosphere may be useful for observing certain features there, e.g. gravity waves, with more detail.*

*The authors claim a cost advantage of the new system launched at relatively few stations compared to maintaining or even enhancing the relatively dense Chinese radiosonde observation network for targeting if severe weather is approaching. While it is encouraging that the data of the new observing system have already been assimilated by weather forecast models in China, the impact on forecasts has been relatively weak but seemingly consistent. The papers referenced in this context (e.g. Wang et al. 2023) are in Chinese and thus impossible for me to follow. I did not try to use automatic translation for this. The results are based on relatively short validation periods (30 days) or on a case study.*

**Response:** We thank Referee #1 (RC1) for her/his positive evaluation of our research. We have addressed RC1's comments and inquires point by point and revised the manuscript carefully.

The RDSS radiosondes of the balloons drifting in the lower stratosphere very useful for observing certain features of gravity waves. Previous studies have conducted gravitational wave inversion using the data of RDSS and achieved excellent results.

Based on the structure function and singular measure relationships, we quantify stratospheric small-scale gravity waves (SGWs) over China, using the Hurst and intermittency parameters, and discuss their relationship with inertia-gravity waves (IGWs). The results show that the enhancement of SGWs in the stratosphere is accompanied by weakening of the IGWs below, which is related to the Kelvin–Helmholtz instability (KHI), and is conducive to the transport of ozone to higher altitudes from lower stratosphere. The parameter space (H1, C1) shows sufficient potential in the analysis of stratospheric disturbances and their role in material transport and energy transfer (Y. He et al.2024).

**REFERENCES:**

He, Y., Zhu, X., Sheng, Z., and He, M.: Identification of stratospheric disturbance information in China based on the round-trip intelligent sounding system, Atmos. Chem. Phys., 24, 3839–3856, https://doi.org/10.5194/acp-24-3839-2024, 2024.

The initial assimilation forecast impact test did indeed show a relatively weak positive contribution, but it was not very significant. This might be because the assimilation techniques for the new RDSS observation data, such as observation error and sparsity schemes, were not optimal. Subsequently, we will further improve the observation error and sparsity schemes in order to obtain better analysis results.

At present, the assimilation experiment has only been completed for one month, which indeed cannot fully demonstrate its effect. Combining the optimization of observation errors and sparsity schemes, we are currently studying a comprehensive optimization plan. We plan to optimize the assimilation parameters such as observation error and sparsity before conducting a one-year batch trial.

*Overall I do not consider the results presented as rigorous proof that the additional measurement data from the drifting balloons improve the quality of analyses and forecasts. While the figures generally support the statements in the text their technical quality is partly poor and should be improved.*

*This leads to the following assessment*

*Scientific significance: fair*

*Scientific quality: fair*

*Presentation quality: fair*

**Response:** We sincerely thank the RC1 for their valuable feedback that we used to improve the quality of the manuscript. Based on the comments and suggestions from the RC1, we have carefully and substantially revised the manuscript. We have included a detailed response addressing each of the comments. The RC1's comments are laid out below in italicized font and specific concerns have been numbered. Our response is given in normal font and changes/additions to the manuscript are given in the blue text. The line numbers correspond to the revised manuscript without the track changes displayed. For precise details on the modifications made in the latest version, please consult the supplementary material, where track changes are enabled.

*2. Major Comments:*

*(1) Some figures are practically unreadable, please redraw or omit:*

**Response:** We thank the RC1 for the valuable and creative comments. Your statement is correct, Thank you for highlighting this issue. I've identified the problematic figures ([list them if possible, e.g. Fig. 4(a)(b), Fig. 6(a)(b)(c), Fig. 7, Fig. 8(a)(b)(c), Fig. 9(a)(b), Fig. 11(a), Fig. 12(a)(b)]) and redraw/omit them with clearer versions by [time/date]. Appreciate your patience!

1) *Fig. 12 is unreadable.*

**Response:** Fig. 12 (a)(b) is a screenshot of an online prediction trajectory display system we established, with a relatively low resolution. Here, we have been omit it. And we have been delete "L471 (Fig. 12a and 12b)" in the revised manuscript.

2) *Fig. 11a): Please use thicker lines or make figure sharper. Red color is used for both showing the simulated vertical speed and the circle for highlighting the descent phase. This is confusing.*

**Response:** Thank you for your opinion. Based on your suggestion, we have redrawn Figure 11(a).

[Figure]

Figure 11. (a) Comparison of simulated (red line) and observed (blue line) vertical speeds of RDSS radiosonde data during the descent phase at the Anqing station at 11:17 on June 20, 2018; (b) Deviations of 63 pairs of simulated RDSS trajectories versus observed trajectories (black line), with the average deviation indicated by the blue line and the largest forecast deviation shown by the red line.

3) *Fig. 9: What do the dots exactly mean? For me it would be logical if red is the start of the ascent phase, green is the start of the drift phase and blue is the start of the descent phase. The end of the black lines would then be the location where the payload reaches the surface. However this is not consistent with how the colors are labelled.*

**Response:** Thank you for your suggestion. Indeed, Figure 9 is not very clear and definite, and "L391-393 The drifting trajectories of RDSS radiosondes in the middle and lower reaches of the Yangtze River in China from July to August 2021 (Fig. 9) uniformly followed a west-to-east path, aligning with the region's strong convective direction."causing confusion for you and the readers. We deleted Figure 9(a) (b) and L391-393. Thank you again for your suggestion, which makes this picture more readable.

4) *Fig. 8: The T-logp diagrams are almost unreadable and they also have different scales. For a publication in a serious journal these must be redrawn.*

**Response:** Thank you for your suggestion. Because these pictures are screenshots from our weather forecast operation system software, they are indeed not clear. Therefore, we redrew the T-logP plot (see Figure 8(a)(b)(c)) and unified the scale in the revised manuscript. And we **modify** "L402-403 Figure 8. Comparison of RDSS GTH3 and GTS1 radiosonde T-logP at the Wuhan station: (a) Wuhan Balloon Sounding at 19:15; (b) Wuhan RDSS ascent phase at 20:00; (c) Wuhan RDSS descent phase at 21:30." **to** "Figure 8. Comparison of RDSS and GTS1 radiosonde T-logP at the Wuhan station: (a) GTS1 operational sounding; (b) ascent phase of RDSS sounding; (c) descent phase of RDSS sounding."

[Figure]

[Figure]

[Figure]

Figure 8. Comparison of RDSS and GTS1 operational soundingT-logP at the Wuhan station: (a) GTS1 operational sounding; (b) ascent phase of RDSS sounding; (c) descent phase of RDSS sounding.

5)  *Fig. 7: Is it possible to zoom in? Most of the information East of Wuhan appears unimportant. The red writing in the chart is unreadable. Should it be "Wuhan"?*

**Response:** Thank you for your suggestion. We apologize for the inconvenience caused to you. The Fig.7 only shows one precipitation weather process. Deleting this picture not affect our discussion. So, we omit this picture and "L400 Figure 7. Combined Reflectivity Factor of Radar Mosaic in the Yangtze River Basin, China, at 20:00 on July 8, 2021."modify "L375 June 8 (Fig. 7) to July 8" in the revised manuscript.

6)  *Fig. 6: The 3D-plot is not helpful. Is it possible to draw the same trajectory information into panel b), using a multi-colored polygon with the color scheme indicating the height of the balloon?*

**Response:** Thank you for your constructive comments. We have plotted the trajectories on a two-dimensional map (Fig. 6) in the revised manuscript. The simulated and observed trajectories are represented by red and black, respectively. A color gradient based on pressure altitude is used to indicate the variation of trajectory height along the path. The Fig.6(b)(c) only shows Temperature and Wind field at 200 hPa in the middle and lower reaches of the Yangtze River. Deleting this picture not affect our discussion. So, we omit this picture in the revised manuscript. And we modify "L352 Figure 6. (a) RDSS three-dimensional trajectory diagram; (b) Temperature field at 200 hPa in the middle and lower reaches of the Yangtze River; (c)

Wind field at 200 hPa in the middle and lower reaches of the Yangtze River."to "Figure.6 Observation (black triangles) and simulation (red dots) trajectory diagram. The yellow pentagrams represent sounding stations, and the colour of the dots represent the corresponding pressure heights. The colour range from light to dark, indicating the process of the trajectory rising from low altitude (high pressure) to high altitude (low pressure)." in revised manuscript.

[Figure]

Figure.6 Observation (black triangles) and simulation (red dots) trajectory diagram. The yellow pentagrams represent sounding stations, and the colour of the dots represent the corresponding pressure heights. The colour range from light to dark, indicating the process of the trajectory rising from low altitude (high pressure) to high altitude (low pressure).

7) *Fig. 4: Same suggestion as for Fig. 6, draw the info of panel b) into panel a). While looking fancy, the extreme exaggeration of the vertical coordinate compared to the horizontal ones is somewhat misleading.*

**Response:** Thank you for your constructive suggestions. We apologize for the inconvenience caused to you. We have plotted the trajectories on a three-dimensional map (Fig. 4) and we modify L333-335 Fig. 4 caption in the revised manuscript.

[Figure]

Figure 4 (a) The network distribution of RDSS and (b) an example of measurement: the trajectory for RDSS at the Anqing station at 12:00 UTC on 11 July 2021. The black triangles represent launch stations of RDSS , while the black dots represent Receive stations of RDSS.

**(2)The added value of the RDSS compared to a sounding system that has just an ascent and a descent phase is unclear and appears limited. ECMWF reported about successfully assimilating descent data**
*https://www.ecmwf.int/sites/default/files/elibrary/102021/20225-newsletter-no-169-autumn-2021_1.pdf, similar to what is reported in this manuscript. There is very little information about the quality of data collected during the drift phase, except for the caveat that radiation errors may be larger than during ascent and descent due to lacking motion of the balloon relative to the atmosphere. However I wonder if the information during the drift may be useful for analyzing gravity wave activity, for example. This aspect is getting increasing attention. The authors appear to see the drift phase more as an opportunity to steer the balloon, less as a measurement period. However the chance to steer the balloon depends a lot on favorable wind conditions. So for targeting one needs to launch the balloon at the right position so that it can descend later into a weather system of interest. Wouldn't that be possible also with conventional radiosondes? The question is really if the money invested into the drifting capability would not be better invested into e.g. balloons that can reach as high up as possible (e.g.*
*https://www.researchgate.net/publication/384155227_Seasonal_and_geographic_via*

*bility_of_high_altitude_balloon_navigation)* or into better humidity sensors that can measure reliably under cold low-pressure conditions.

**Response:** We read the technical report about ECMWF successfully assimilating descent data carefully and were greatly inspired. We fully agree with the RC1's insights on the value of radiosonde descent data. As highlighted by recent operational implementations (ECMWF, 2021), carefully processed descent profiles from Vaisala RS41 radiosondes provide scientifically valid atmospheric data under specific conditions: Exclusion of data ≤150 hPa to mitigate high-speed descent errors; Mandatory pressure sensors for tropospheric pressure correction; Prioritization of ocean-derived profiles due to their unique ability to capture near-surface dynamics in data-sparse regions.

We fully agree that descent-phase observations are of great importance. By extending the drifting phase in RDSS, we are not only able to obtain long-duration observations in the lower stratosphere but also gain several additional benefits beyond those you pointed out, such as their application in gravity wave studies. The advantages of incorporating a longer drifting phase are as follows:

1) It provides a wealth of long-duration observational data in the lower stratosphere, which can be used for refined verification of stratospheric forecasts in numerical models.

2) With the inclusion of an extended drifting phase, the descent-phase observations often occur during the traditional radiosonde data gap periods (i.e., 06 and 18 UTC compared to the standard 00 and 12 UTC times), thus offering valuable supplementary data.

3) The addition of a drifting phase results in descent-phase observations occurring far from the launch site. This enables the acquisition of vertical atmospheric profiles in regions that are otherwise sparsely observed, such as the margins of the Tibetan Plateau or over the ocean near coastal stations. These observations provide unique, direct vertical data for numerical modeling and weather research that have not been previously available.

We have already utilized the RDSS trajectory simulation system to model the spatial distribution of descent-phase observation locations for 120 radiosonde stations across China. The results are shown in Figure R1, where different colors represent different months. Compared with traditional sounding observations, which are limited to the areas near sounding stations, the additional descent phase of the RDSS enables atmospheric vertical profile observations at virtually any location nationwide. This capability effectively supplements the atmospheric vertical profile data in regions far from sounding stations, sparsely distributed areas (e.g., the Tibetan Plateau and its surroundings), and regions where it is difficult to set up sounding stations (e.g., vast uninhabited regions of the Tibetan Plateau and the oceans). Once the nationwide RDSS network is established, it will provide extensive atmospheric vertical profile observations for regions where such data are currently scarce, thereby supporting the NWP and atmospheric science research. This advancement will be of great significance for improving the understanding of weather patterns in China and enhancing the accuracy of the NWP.

[Figure]

Fig. R1 Spatial distribution of the landing points (colored dots) for sounding stations (black triangles) across China from February 2022 to January 2024. Different colors represent different months.

***(3) There is little information about availability of these data to the public. They are potentially valuable for weather centers around the globe or for atmospheric climate reanalysis. It would be important for the reader to know whether these data can be accessed and used under a general public license.***

**Response:** The field experiments data of RDSS involves data policies and can only be shared to a limited extent. It can be shared on a small scale through cooperation between both parties. The CMA is actively studying and formulating the sharing strategy. The authors will actively strive to promote the sharing of this data.

***3.Minor Comments:***

*1) Table 3: Please explain what "Airspeed" means. Is it the data transmission rate in bits per second?*

**Response:** Yes, Thank you for your rigorous review. There is a word error.The "Airspeed" meaning is not clear, we will replace the "Data Transmission Rate" in revised manuscript.

*2) L295: It is mentioned that the effect of radiation in the drift phase is greater than during ascent/descent. Can you quantify that, ideally with a plot or table showing the*

*increased measurement uncertainty during the drift phase. It is unclear to the reader how the CFD correction model works and how it performs.*

**Response:** Thank you for your constructive suggestions. We will draw a figure of "decreased effect of radiation uncertainty during the drift phase" in revised manuscript.

[Figure]

Fig. R2 decreased effect of radiation error during the ADD phase of RDSS.

*3)Table 4.2: The assessment is rather crude, particularly for humidity. Having just one value for the whole troposphere is insufficient by today's standard. Can you give a profile of the humidity measurement uncertainty?*

**Response:** Thank you for your constructive suggestions. We have another manuscript is under review:

Beidou Navigation Radiosonde Observation Experiment and Data Evaluation, Lebao Yao, Dan Shen, Xin Sun, Donghai Wang, Xiaozhong Cao, Jincheng Wang, Dan Wang, Chunyan Zhang, Qiyun Guo. Submitted to Atmospheric Research

which is assessment "profile of the humidity measurement uncertainty"

*4)L368: track the occurrence of the entire convective system and the changes .. in real time... is a bit strong wording, given the fine grained structure visible in the RADAR picture which is by no means resolvable with a few radiosondes, even with drifting capability.*

**Response:** We will replace the strong wording ".. track the occurrence of the entire convective system and the changes .. in real time..." in revised manuscript.

*5)L425: results from one month are not conclusive. There should be at least three months (e.g. used as minimum by ECMWF when upgrading their system), ideally from different seasons, to cover any dependency on the annual cycle.*

**Response:** Sorry, At present, the assimilation experiment has only been completed for one month, which indeed cannot fully demonstrate its effect. Combining the optimization of observation errors and sparsity schemes, we are currently studying a comprehensive optimization plan. We plan to optimize the assimilation parameters such as observation error and sparsity before conducting a one-year batch trial.

*6)L534: The reduction of the forecast trajectory error in one typhoon event must be considered anecdotal. While encouraging it is certainly not science-grade evidence of an improvement.*

**Response:** Thank you for your constructive suggestions. The science-grade evidence of an improvement is a long process. But we will make persistent efforts to prove it. And we cite relevant paper in revised manuscript.

**REFERENCES:**

Qiushi Wen, Xuefen Zhang, Sheng Hu, et,al. Collaborative assimilation experiment of Beidou radiosonde and drone-dropped radiosonde based on CMA-TRAMS[J]. Atmospheric and Oceanic Science Letters, 2025, 18(2). doi:10.1016/j.aosl.2024.100555.

*7)L550: It is not obvious from the material presented that the RDSS really reduces costs compared to conventional radiosondes. This needs more detailed explanation.*

**Response:** In fact, one RDSS can provide 'Ascent-Drift-Descent' three-phase sounding in which all three phases of sounding observation are executed through single balloon launch. Compared to conventional (Ascent) soundings, it adds 'drift-descent' soundings. So, we modify "L550 Compared to intensive soundings,"to "Compared to two of conventional soundings," in revised manuscript.

*8)L567: This statement on data availability is unacceptable by today's open science standards.*

**Response:** The field experiments data of RDSS involves data policies and can only be shared to a limited extent. It can be shared on a small scale through cooperation between both parties. And we modify "L567 Requests for data that support the findings of this study can be sent to luohw_1@163.com."to "The CMA is actively studying and formulating the sharing strategy. The authors will actively strive to promote the sharing of this data. According to the relevant policies of the CMA, this data currently cannot be widely shared. For data sharing, you can contact us to discuss cooperation and data sharing." in revised manuscript.

---

## Community Comment (CC3)

**Reply TO 'Comment on egusphere-2025-2012', Anonymous Referee #2, 16 Jul 2025**

**Response to Referee #2's comments**

*Referee #2: egusphere-2025-2012*

*Synopsis: The Round-trip Drifting Sounding System (RDSS) proposed in this study demonstrates significant innovation and holds promising application potential. It is recommended for publication after minor revision. The RDSS's key advantage lies in completing a continuous atmospheric observation cycle of "ascent (1 hour) - drifting (4 hours) - descent (1 hour)" with a single balloon launch. Notably, the drifting phase provides continuous observation of the lower stratosphere, while the descent phase provides atmospheric vertical profiles over remote areas far from the launch site. This effectively addresses the gap in traditional sounding observations at 06 and 18 UTC. Furthermore, by utilizing long-distance drifting, RDSS expands the capability for vertical atmospheric sounding in remote regions through the descent observations.*

*Field experiments conducted in the Yangtze River Basin successfully tackled several key technical challenges. Compared to China's previous-generation L-band sounding system, RDSS exhibits reduced observation errors, particularly in tropospheric wind fields and humidity. Additionally, RDSS possesses a degree of mobility, suggesting potential for targeted observation campaigns. The study also presents preliminary results from RDSS data assimilation and forecast impact experiments, indicating a significant improvement in the skill of 12-24 hour accumulated precipitation forecasts initialized at 06UTC and 18UTC.*

**Response:** We thank Referee #2 (RC2) for her/his positive evaluation of our research. We have addressed each of reviewer's comments and inquires point by point and revised the manuscript carefully. And We sincerely thank the Referee #2 (RC2) for her/his valuable feedback that we used to improve the quality of the manuscript. Based on the comments and suggestions from the RC2, we have carefully and substantially revised the manuscript. We have included a detailed response addressing each of comments. The comments are laid out below in italicized font and specific concerns have been numbered. Our response is given in normal font and changes/additions to the manuscript are given in the blue text. The line numbers correspond to the revised manuscript without the track changes displayed. For precise

details on the modifications made in the latest version, please consult the supplementary material, where track changes are enabled.

***Specific Recommendations for Revision:***

1. *Table 1: The horizontal lines in the three-line table vary in thickness. It is recommended to standardize the line weight.*

**Response:** We thank the RC2 for the valuable and creative comments. Your statement is correct, we standardize the line weight of Table 1 in revised manuscript.

| No | | Instruments | Key Function |
|---|---|---|---|
| 1 | "ADD" subsystem | zero-pressure dual-mode meteorological balloon | "outer balloon"as ascent carrier,"inner balloon" as drift carrier |
| 2 | | parachute | parachute as the carrier of the descent phase |
| 3 | | drifting controller | Adaptive control of drift and descent |
| 4 | | radiosonde | The temperature, pressure, humidity, wind measurement meet the demand for long-term stratospheric observation |
| 5 | Ground operation control subsystem | ground station | ground inspection ground check, balloon inflation, launch, and other tasks before the equipment is launched |
| 6 | | ground data-receiving device | 8 channels receive radiosonde data simultaneously |
| 7 | | control command transmitter | In the weather-sensitive area without a station, the active fusing drifting controller is carried out and the descent measurement is started |
| 8 | | operational management system | Real-time acquisition, transmission, quality control, and timely delivery of control instructions for RDSS data, providing real-time high-quality data to weather analysis and numerical prediction models |

2. *Table 4: Words are hyphenated across lines. It is recommended to adjust the table format to prevent word breaks.*

**Response:** We thank the RC2 for the valuable and creative comments. Your statement is correct, we prevent word breaks,the new Table 4 in revised manuscript.

| Time | Height | Atmospheric temperature [K] | Relative humidity [%RH] | Geopotential height [m] | Pressure [hPa] | Wind (horizontal)direction [°] | Wind (horizontal)speed [ms⁻¹] | Wind (horizontal)vector [ms⁻¹] |
|---|---|---|---|---|---|---|---|---|
| Day | PBL | $0.18_{0.17}^{-0.05}\pm0.03$ | $7.00_{4.41}^{-5.43}\pm0.74$ | X | X | X | X | X |
| | FT | $0.12_{0.11}^{+0.05}\pm0.04$ | $8.75_{8.02}^{-3.50}\pm0.60$ | $5.9_{5.5}^{+2.0}\pm1.8$ | $0.4_{0.4}^{-0.0}\pm0.1$ | $3.6_{3.6}^{-0.4}\pm0.2$ | $0.2_{0.2}^{-0.0}\pm0.0$ | $0.3_{0.1}^{+0.2}\pm0.0$ |
| | UTLS | $0.09_{0.08}^{+0.01}\pm0.03$ | $7.73_{7.58}^{-1.55}\pm0.40$ | $13.2_{8.6}^{+10.0}\pm3.8$ | $0.4_{0.2}^{-0.3}\pm0.1$ | $2.5_{2.5}^{-0.2}\pm0.3$ | $0.2_{0.2}^{-0.0}\pm0.0$ | $0.3_{0.2}^{+0.2}\pm0.0$ |
| | MUS | $0.27_{0.16}^{-0.22}\pm0.10$ | $1.69_{0.82}^{+1.48}\pm0.46$ | $29.5_{17.9}^{+23.4}\pm4.2$ | $0.3_{0.1}^{-0.2}\pm0.0$ | $6.1_{6.1}^{-0.4}\pm0.2$ | $1.3_{1.3}^{-0.0}\pm0.0$ | $1.5_{1.5}^{+0.3}\pm0.0$ |
| Night | PBL | $0.38_{0.34}^{-0.18}\pm0.05$ | $4.72_{4.66}^{+0.74}\pm0.15$ | X | X | X | X | X |
| | FT | $0.15_{0.15}^{+0.02}\pm0.02$ | $6.41_{6.03}^{+2.16}\pm0.11$ | $5.8_{5.8}^{+0.4}\pm0.4$ | $0.5_{0.5}^{+0.1}\pm0.2$ | $2.6_{2.6}^{-0.2}\pm0.2$ | $0.2_{0.2}^{-0.0}\pm0.0$ | $0.2_{0.1}^{+0.2}\pm0.0$ |
| | UTLS | $0.12_{0.10}^{+0.06}\pm0.05$ | $6.82_{5.74}^{+3.70}\pm0.26$ | $11.5_{8.6}^{+7.7}\pm3.4$ | $0.3_{0.2}^{-0.1}\pm0.1$ | $2.4_{2.4}^{-0.1}\pm0.1$ | $0.2_{0.2}^{+0.0}\pm0.0$ | $0.2_{0.1}^{+0.2}\pm0.0$ |
| | MUS | $0.10_{0.10}^{-0.03}\pm0.02$ | $1.71_{0.74}^{+1.54}\pm0.28$ | $26.7_{16.8}^{+20.7}\pm4.2$ | $0.1_{0.1}^{-0.1}\pm0.0$ | $4.5_{4.4}^{-0.6}\pm0.2$ | $0.2_{0.2}^{-0.0}\pm0.0$ | $0.4_{0.3}^{+0.3}\pm0.0$ |

3. *Figure 6(a): The 3D plot lacks clarity. It is recommended to replace it with a 2D plot, using different colors to represent trajectory altitudes.*

**Response:** Thank you for your constructive comments. We have plotted the trajectories on a two-dimensional map (Fig. 6) in the revised manuscript. The simulated and observed trajectories are represented by red and black, respectively. A color gradient based on pressure altitude is used to indicate the variation of trajectory height along the path. The Fig.6(b)(c) only shows Temperature and Wind field at 200 hPa in the middle and lower reaches of the Yangtze River. Deleting this picture will not affect our discussion. So, we omit this picture in the revised manuscript.

[Figure]

Fig.6 Observation (black triangles) and simulation (red dots) trajectory diagram. The yellow pentagrams represent sounding stations, and the colors of the dots represent the corresponding pressure heights. The colors range from light to dark, indicating the process of the trajectory rising from low altitude (high pressure) to high altitude (low pressure).

4. *Figure 7: This figure solely displays radar reflectivity for a specific weather event and has weak relevance to the main focus of the paper. It is recommended for deletion.*

**Response:** Thank you for your suggestion. We apologize for the inconvenience caused to you. The Fig.7 only shows one precipitation weather process. Deleting this picture will not affect our discussion. So, we omit this picture and "L374-375Convection developed and moved to northern Jiangxi, northern Zhejiang, southern Anhui, and southern Jiangsu overnight on July 8 (Fig. 7)" "L400 Figure 7. Combined Reflectivity Factor of Radar Mosaic in the Yangtze River Basin, China, at 20:00 on July 8, 2021."in the revised manuscript.

5. *Figures 8(a)-(c): The three T-logP diagrams are unclear. Furthermore, they have differing vertical axis ranges and inconsistent sizes, which is unsuitable for formal journal publication. It is strongly recommended to redraw these figures uniformly.*

**Response:** Thank you for your suggestion. Because these pictures are screenshots from our weather forecast operation system software, they are indeed not clear. Therefore, we redrew the T-logP plot (see Figure 8(a)(b)(c)) and unified the scale in the revised manuscript. And we modify "L402-403 Figure 8. Comparison of RDSS GTH3 and GTS1 radiosonde T-logP at the Wuhan station: (a) Wuhan Balloon Sounding at 19:15; (b) Wuhan RDSS ascent phase at 20:00; (c) Wuhan RDSS descent phase at 21:30." to "Figure 8. Comparison of RDSS and GTS1 radiosonde T-logP at the Wuhan station: (a) GTS1 operational sounding; (b) ascent phase of RDSS sounding; (c) descent phase of RDSS sounding."

[Figure]

[Figure]

StationID:57494(Wuhan)    Time:2021-07-08 19:13:00～20:25:00

| element | obs |
|---|---|
| -20H (m) | 8286.23 |
| ZH (m) | 4733.57 |
| CAPE (J/KG) | 1498.16 |
| CIN (J/KG) | 35.65 |
| PWAT(mm) | 66.73 |
| K | 38.13 |
| LFC (hPa) | 712.57 |
| LCL (hPa) | 945.22 |

[Figure]

Figure 8. Comparison of RDSS and GTS1 operational soundingT-logP at the Wuhan station: (a) GTS1 operational sounding; (b) ascent phase of RDSS sounding; (c) descent phase of RDSS sounding.

6. *Figure 9:*

   ■ *(a): This panel merely illustrates the experimental domain and provides limited information. It is recommended for deletion; the location can be described textually in the main body.*

   ■ *(b): The symbols are confusing and undefined. It is recommended to recreate this figure with clear labeling of all symbols in the legend. The caption should provide a detailed explanation of the figure's content.*

**Response:** Thank you for your suggestion. Indeed, Figure 9 is not very clear and definite, and "L391-393 The drifting trajectories of RDSS radiosondes in the middle and lower reaches of the Yangtze River in China from July to August 2021 (Fig. 9) uniformly followed a west-to-east path, aligning with the region's strong convective direction."causing confusion for you and the readers. We deleted Figure 9(a) (b) and L391-393.Thank you again for your suggestion, which makes this picture more readable.

7. *Figure 12: The image is unclear and primarily demonstrates the trajectory visualization software's output. It has minimal connection to the paper's core innovations and technical content. It is recommended for deletion.*

**Response:** Fig. 12 is a screenshot of an online prediction trajectory display system we established, with a relatively low resolution. Here, we will delete it.

8. ***Data Assimilation Experiments:*** *The assimilation application experiments using RDSS data in this paper are relatively limited. It is recommended that subsequent research focuses on: (a) Further refining assimilation techniques for RDSS second-level data (e.g., data thinning methods); (b) Conducting more extensive numerical forecast assimilation impact experiments utilizing additional observational data; (c) Conducting a thorough evaluation of the forecast skill improvement offered by RDSS compared to China's conventional L-band sounding system.*

**Response:**Thank you for the suggestions of the RC2. In the subsequent research, we will evaluate and improve the data on numerical forecasting for the RDSS about your suggestion.

9. *"Zhuang Z R, Wang R C (2019), Wang J C, et al." - Literature duplication.*

**Response:**Thank you for the suggestions of the RC2. We apologize for the inconvenience caused to you. We will delate L715-716 in revised manuscript.

10. *The two attached papers also studied and confirmed the impact and value of the RDSS data on numerical forecasting. It is recommended to include them.*

*ZHANG Xin, WANG Qiuping, MA Xulin, et al. 2025. The Influence of New Round-Trip Drifting Sounding Observation on the Quality of Numerical Prediction in the Middle and Lower Reaches of the Yangtze River [J]. Chinese Journal of Atmospheric Sciences, 49(1): 245−256. doi:10.3878/j.issn.1006-9895.2304.22224*

*Zhang, X., Sun, L., Ma, X., Guo, H., Gong, Z., Yan, X. Can the Assimilation of the Ascending and Descending Sections' Data from Round-Trip Drifting Soundings Improve the Forecasting of Rainstorms in Eastern China? Atmosphere 2023, 14, 1127. https://doi.org/10.3390/ atmos14071127*

**Response:** We read the two papers. It was of great help to us. In the subsequent research, And we will evaluate and improve the data on numerical forecasting for the application of RDSS.

**We will add the two paper attached in REFERENCES:**

ZHANG Xin, WANG Qiuping, MA Xulin, et al. 2025. The Influence of New Round-Trip Drifting Sounding Observation on the Quality of Numerical Prediction in the Middle and Lower Reaches of the Yangtze River [J]. Chinese Journal of Atmospheric Sciences, 49(1): 245−256. doi:10.3878/j.issn.1006-9895.2304.22224

Zhang, X., Sun, L., Ma, X., Guo, H., Gong, Z., Yan, X. Can the Assimilation of the Ascending and Descending Sections' Data from Round-Trip Drifting Soundings

Improve the Forecasting of Rainstorms in Eastern China? Atmosphere 2023, 14, 1127. https://doi.org/10.3390/ atmos14071127

Improve the Forecasting of Rainstorms in Eastern China? Atmosphere 2023, 14, 1127. https://doi.org/10.3390/ atmos14071127

---

## Author Comment (AC1)

We thank the three reviewers for their valuable suggestions and constructive criticism, which helped improve our analysis and manuscript. Below we respond to all referee comments (RCs) provide a detailed point-by-point response (in Blue purple) to the reviewers' comments (in Black).

**Response to Referee #1's comments**

*Synopsis: The manuscript describes a novel upper air balloon observing system, consisting of a double balloon setup that allows for measurements during a long drift phase in the stratosphere and during the descent phase. The start of the drift phase and of the descent phase can be triggered remotely and steering the height is possible as well. The manuscript also describes the potential for using the system for targeting observations and how assimilation of the data improves analysis and forecasts.*

*1.Overall comment: It is certainly desirable to make better use of weather balloons than is currently the case with many conventional radiosondes where only data collected during the ascent phase are used. The longer residence of the balloons in the lower stratosphere may be useful for observing certain features there, e.g. gravity waves, with more detail.*

*The authors claim a cost advantage of the new system launched at relatively few stations compared to maintaining or even enhancing the relatively dense Chinese radiosonde observation network for targeting if severe weather is approaching. While it is encouraging that the data of the new observing system have already been assimilated by weather forecast models in China, the impact on forecasts has been relatively weak but seemingly consistent. The papers referenced in this context (e.g. Wang et al. 2023) are in Chinese and thus impossible for me to follow. I did not try to use automatic translation for this. The results are based on relatively short validation periods (30 days) or on a case study.*

**Authors: -** We thank Referee #1 (RC1) for her/his positive evaluation of our research. We have addressed RC1's comments and inquires point by point and revised the manuscript carefully.

The ADDRS radiosondes of the balloons drifting in the lower stratosphere very useful for observing certain features of gravity waves. Previous studies have conducted gravitational wave inversion using the data of ADDRS and achieved excellent results. Based on the structure function and singular measure relationships, we quantify stratospheric small-scale gravity waves (SGWs) over China, using the Hurst and intermittency parameters, and discuss their relationship with inertia-gravity waves

(IGWs). The results show that the enhancement of SGWs in the stratosphere is accompanied by weakening of the IGWs below, which is related to the Kelvin–Helmholtz instability (KHI), and is conducive to the transport of ozone to higher altitudes from lower stratosphere. The parameter space (H1, C1) shows sufficient potential in the analysis of stratospheric disturbances and their role in material transport and energy transfer (Y. He et al.2024).

**REFERENCES:**

He, Y., Zhu, X., Sheng, Z., and He, M.: Identification of stratospheric disturbance information in China based on the round-trip intelligent sounding system, Atmos. Chem. Phys., 24, 3839–3856, https://doi.org/10.5194/acp-24-3839-2024, 2024.

The initial assimilation forecast impact test did indeed show a relatively weak positive contribution, but it was not very significant. This might be because the assimilation techniques for the new ADDRS observation data, such as observation error and sparsity schemes, were not optimal. Subsequently, we will further improve the observation error and sparsity schemes in order to obtain better analysis results.

At present, the assimilation experiment has only been completed for one month, which indeed cannot fully demonstrate its effect. Combining the optimization of observation errors and sparsity schemes, we are currently studying a comprehensive optimization plan. We plan to optimize the assimilation parameters such as observation error and sparsity before conducting a one-year batch trial.

*Overall I do not consider the results presented as rigorous proof that the additional measurement data from the drifting balloons improve the quality of analyses and forecasts. While the Figs generally support the statements in the text their technical quality is partly poor and should be improved.*

*This leads to the following assessment*

*Scientific significance: fair*

*Scientific quality: fair*

*Presentation quality: fair*

**Authors: -** We sincerely thank the RC1 for their valuable feedback that we used to improve the quality of the manuscript. Based on the comments and suggestions from the RC1, we have carefully and substantially revised the manuscript. We have

included a detailed response addressing each of the comments. The RC1's comments are laid out below in italicized font and specific concerns have been numbered. Our response is given in normal font and changes/additions to the manuscript are given in the blue text. The line numbers correspond to the revised manuscript without the track changes displayed. For precise details on the modifications made in the latest version, please consult the supplementary material, where track changes are enabled.

*2.Major Comments:*

*(1)Some Figs are practically unreadable, please redraw or omit:*

**Authors: -** We thank the RC1 for the valuable and creative comments. Your statement is correct, Thank you for highlighting this issue. I've identified the problematic Figs ( Fig. 4(a)(b), Fig. 6(a)(b)(c), Fig. 7, Fig. 8(a)(b)(c), Fig. 9(a)(b), Fig. 11(a), Fig. 12(a)(b)) and redraw/omit them with clearer versions. Appreciate your patience!

1)  *RC1:Fig. 12 is unreadable.*

*RC2:Fig 12: The image is unclear and primarily demonstrates the trajectory visualization software's output. It has minimal connection to the paper's core innovations and technical content. It is recommended for deletion.*

**Authors: -** Thank you to the two reviewers for your insightful views. Fig. 12 (a)(b) is a screenshot of an online prediction trajectory display system, It has minimal connection to the paper's core innovations and technical content. Here, we have been omit it in revised manuscript.

2)  *Fig. 11a): Please use thicker lines or make Fig sharper. Red color is used for both showing the simulated vertical speed and the circle for highlighting the descent phase. This is confusing.*

**Authors: -** Thank you for your opinion. Based on your suggestion, we have redrawn Fig 11(a) in line 491-495 of the revised manuscript.

[Figure]

Fig 11. (a) Comparison of simulated (red line) and observed (blue line) vertical speeds of ADDRS radiosonde data during the descent phase at the Anqing station at 11:17 on June 20, 2018; (b) Deviations of 63 pairs of simulated ADDRS trajectories versus observed trajectories (black line), with the average deviation indicated by the blue line and the largest forecast deviation shown by the red line.

3) *RC1:Fig. 9: What do the dots exactly mean? For me it would be logical if red is the start of the ascent phase, green is the start of the drift phase and blue is the start of the descent phase. The end of the black lines would then be the location where the payload reaches the surface. However this is not consistent with how the colors are labelled.*

*RC2:Fig 9:(a): This panel merely illustrates the experimental domain and provides limited information. It is recommended for deletion; the location can be described textually in the main body.(b): The symbols are confusing and undefined. It is recommended to recreate this Fig with clear labeling of all symbols in the legend. The caption should provide a detailed explanation of the Fig's content.*

*RC3:Fig 9b. A lot of green 'descent' dots are close to the 'ascent' point. I wondered if green is actually the start of the drift phase and blue the start of descent? Some people are red-green colour blind, changing one of the colors would help them. (Unlike reviewer 2 I find the orientation given by Fig 9a useful.)*

**Authors: -** Yes, RC1 are corrected. "red is the start of the ascent phase, green is the start of the drift phase and blue is the start of the descent phase. The end of the black lines would then be the location where the payload reaches the surface." but sadly Fig 9 was basically drifting from east to west, which contradicts the original text "L391-393 The drifting trajectories of ADDRS radiosondes in the middle and lower reaches of the Yangtze River in China from July to August 2021 (Fig. R1) uniformly followed a west-to-east path, aligning with the region's strong convective direction." It is the PBL planetary wind belt near 30° north latitude in the middle and lower reaches of the Yangtze River in China. The airflow emitted from the subtropical high to the north polar low is deflected into westerly winds under the action of the Coriolis

effect force. Therefore, the strong convection within the troposphere develops from west to east along with the planetary wind belt. In the mid-latitudes of the Northern hemisphere, the stratosphere is dominated by easterly winds in summer, so the radiosonde will move from east to west drifting in the stratosphere.

Fig.R2 as shown in total 8 groups,among the 4 groups that successfully drifted were all from east to west drifting, while the 4 groups that failed to drift were observed from west to east within the troposphere. Therefore, in the mid-latitudes of China during summer, the stratospheric drift is in the direction from east to west, which is opposite to the strong convective direction from west to east.

However, since all summer typhoons are formed in the southeast Pacific and make landfall in the Chinese mainland, the movement trajectory of the drift observation can be well used for targeted observation of typhoons.

So, we deleted Fig 9(a) (b) and L391-393. Thank you again for your suggestion, which makes this section more reasonable.

[Figure]

**Fig R1(Fig 9(a) (b)). The ADDRS drifting trajectory from July to August 2021.**

[Figure]

**Fig.R2 On July 8th and 9th at 00 and 12UTC in 2021, the trajectories of a total of 8 Beidou radiosondes in the Anqing and Wuhan radiosonde observation stations.**

4)  *RC1:Fig. 8: The T-logp diagrams are almost unreadable and they also have different scales. For a publication in a serious journal these must be redrawn.*

*RC2:Figs 8(a)-(c): The three T-logP diagrams are unclear. Furthermore, they have differing vertical axis ranges and inconsistent sizes, which is unsuitable for formal journal publication. It is strongly recommended to redraw these Figs uniformly.*

**Authors: -** Thank you for your suggestion. Because these pictures are screenshots from our weather forecast operation system software, they are indeed not clear. Therefore, we redrew the T-logP plot (see Fig 8(a)(b)(c)) and unified the scale on Page 17 , L416 of the revised manuscript. And we **modify** "L402-403 Fig 8. Comparison of ADDRS GTH3 and GTS1 radiosonde T-logP at the Wuhan station: (a) Wuhan Balloon Sounding at 19:15; (b) Wuhan ADDRS ascent phase at 20:00; (c) Wuhan ADDRS descent phase at 21:30." **to** "Fig 8. Comparison of ADDRS and GTS1 radiosonde T-logP at the Wuhan station: (a) GTS1 operational sounding; (b) ascent phase of ADDRS sounding; (c) descent phase of ADDRS sounding." in the L417-419 of revised manuscript..

[Figure]

**Fig 9. Comparison of ADDRS and GTS1 operational radiosonde T-logP at the Wuhan station: (a) GTS1 operational radiosonde; (b) ascent phase of ADDRS radiosonde; (c) descent phase of ADDRS radiosonde.**

5) *RC1:Fig. 7: Is it possible to zoom in? Most of the information East of Wuhan appears unimportant. The red writing in the chart is unreadable. Should it be "Wuhan"?*

*RC2:Fig 7: This Fig solely displays radar reflectivity for a specific weather event and has weak relevance to the main focus of the paper. It is recommended for deletion.*

*RC3:Fig 7. Wuhan should be better marked (there is something written in red that I can't read). Perhaps a coloured X or other symbol, described in the caption would be better.*

**Authors: -** Thank you for your suggestion. We apologize for the inconvenience caused to you. We redraw this picture and "L400 Fig 7. Combined Reflectivity Factor of Radar Mosaic in the Yangtze River Basin, China, at 20:00 on July 8, 2021."modify to "Fig.8 The trajectory for GTH2 radiosonde and GTS1 radiosonde at the Wuhan station and radar reflectivity image at 12:00 UTC on 11 July 2021." in the L394-396 of revised manuscript.Fig.8 presents the trajectory data for both the ADDRS radiosonde and the GTS1 radiosonde, which were recorded at red triangle Wuhan radiosonde observation launch station.

[Figure]

**Fig.8 The trajectory for ADDRS radiosonde and GTS1 radiosonde at the Wuhan radiosonde observation launch station and radar reflectivity image at 12:00 UTC on 11 July 2021.**

6) *RC1:Fig. 6: The 3D-plot is not helpful. Is it possible to draw the same trajectory information into panel b), using a multi-colored polygon with the color scheme indicating the height of the balloon?*

*RC2:Fig 6(a): The 3D plot lacks clarity. It is recommended to replace it with a 2D plot, using different colors to represent trajectory altitudes.*

**Authors: -** Thank you for your constructive comments. We have plotted the trajectories on a two-dimensional map (Fig. 7) on Page 15, L365 of the revised manuscript. The simulated and observed trajectories are represented by red and black, respectively. A color gradient based on pressure altitude is used to indicate the variation of trajectory height along the path. The Fig.6(b)(c) only shows Temperature and Wind field at 200 hPa in the middle and lower reaches of the Yangtze River. Deleting this picture not affect our discussion. So, we omit this picture in the revised manuscript.

And we modify "L351-L352 Fig. 6(a) ADDRS three-dimensional trajectory diagram; (b) Temperature field at 200 hPa in the middle and lower reaches of the Yangtze River; (c) Wind field at 200 hPa in the middle and lower reaches of the Yangtze River."to "Fig.6 Observation (black triangles) and simulation (red dots) trajectory diagram. The yellow pentagrams represent sounding stations, and the colour of the dots represent the corresponding pressure heights. The colour range from light to dark, indicating the process of the trajectory rising from low altitude (high pressure) to high altitude (low pressure)." in L366-369 of revised manuscript.

[Figure]

**Fig 7. Observation (black triangles) and simulation (red dots) trajectory diagram. The yellow pentagrams represent sounding stations, and the colour of the dots represent the corresponding pressure heights. The colour range from light to dark, indicating the process of the trajectory rising from low altitude (high pressure) to high altitude (low pressure).**

7) *Fig. 4: Same suggestion as for Fig. 6, draw the info of panel b) into panel a). While looking fancy, the extreme exaggeration of the vertical coordinate compared to the horizontal ones is somewhat misleading.*

**Authors: -** Thank you for your constructive suggestions. We apologize for the inconvenience caused to you. We have plotted the trajectories on a three-dimensional map (Fig. 4) and we modify on Page 13of L335-338 of revised manuscript.

[Figure]

**Fig 4. The network distribution of ADDRS and an example of measurement: the trajectory for ADDRS at the Anqing station at 12:00 UTC on 11 July 2021. The black triangles represent launch stations of ADDRS , while the black dots represent Receive stations of ADDRS.**

*(2)The added value of the RDSS compared to a sounding system that has just an ascent and a descent phase is unclear and appears limited. ECMWF reported about successfully assimilating descent data* [https://www.ecmwf.int/sites/default/files/elibrary/102021/20225-newsletter-no-169-autumn-2021_1.pdf,](https://www.ecmwf.int/sites/default/files/elibrary/102021/20225-newsletter-no-169-autumn-2021_1.pdf) *similar to what is reported in this manuscript. There is very little information about the quality of data collected during the drift phase, except for the caveat that radiation errors may be larger than during ascent and descent due to lacking motion of the balloon relative to the atmosphere. However I wonder if the information during the drift may be useful for analyzing gravity wave activity, for example. This aspect is getting increasing attention. The authors appear to see the drift phase more as an opportunity to steer the balloon, less as a measurement period. However the chance to steer the balloon depends a lot on favorable wind conditions. So for targeting one needs to launch the balloon at the right position so that it can descend later into a weather system of interest. Wouldn't that be possible also with conventional radiosondes? The question is really if the money invested into the drifting capability would not be better invested into e.g. balloons that can reach as high up as possible (e.g.* [https://www.researchgate.net/publication/384155227_Seasonal_and_geographic_viability_of_high_altitude_balloon_navigation)](https://www.researchgate.net/publication/384155227_Seasonal_and_geographic_viability_of_high_altitude_balloon_navigation) *or into better humidity sensors that can measure reliably under cold low-pressure conditions.*

**Authors: -** We read the technical report about ECMWF successfully assimilating descent data carefully and were greatly inspired. We fully agree with the RC1's

insights on the value of radiosonde descent data. And we add "The ECMWF reported about successfully assimilating radiosonde descent data from ships(Ingleby, 2021)," on Page 2, L66-67 of the revised manuscript.

As highlighted by recent operational implementations (ECMWF, 2021), carefully processed descent profiles from Vaisala RS41 radiosondes provide scientifically valid atmospheric data under specific conditions: Exclusion of data ≤150 hPa to mitigate high-speed descent errors; Mandatory pressure sensors for troposphere pressure correction; Prioritization of ocean-derived profiles due to their unique ability to capture near-surface dynamics in data-sparse regions.

We fully agree that descent-phase observations are of great importance. By extending the drifting phase in ADDRS, we are not only able to obtain long-duration observations in the lower stratosphere but also gain several additional benefits beyond those you pointed out, such as their application in gravity wave studies. The advantages of incorporating a longer drifting phase are as follows:

1) It provides a wealth of long-duration observational data in the lower stratosphere, which can be used for refined verification of stratospheric forecasts in numerical models.
2) With the inclusion of an extended drifting phase, the descent-phase observations often occur during the traditional radiosonde data gap periods (i.e., 06 and 18 UTC compared to the standard 00 and 12 UTC times), thus offering valuable supplementary data.
3) The addition of a drifting phase results in descent-phase observations occurring far from the launch site. This enables the acquisition of vertical atmospheric profiles in regions that are otherwise sparsely observed, such as the margins of the Tibetan Plateau or over the ocean near coastal stations. These observations provide unique, direct vertical data for numerical modeling and weather research that have not been previously available.

We have another manuscript is under review:

Manuscript ID ACTA-E-2025-0081.R1 entitled "Scenario Projections for the Round-trip Drifting Sounding System Observations". Submitted to Journal of Meteorological Research (JMR)

We have already utilized the ADDRS trajectory simulation system to model the spatial distribution of descent-phase observation locations for 120 radiosonde stations across China. The results are shown in Fig R3, where different colors represent different months. Compared with traditional sounding observations, which are limited to the areas near sounding stations, the additional descent phase of the ADDRS enables atmospheric vertical profile observations at virtually any location nationwide. This capability effectively supplements the atmospheric vertical profile data in regions far from sounding stations, sparsely distributed areas (e.g., the Tibetan Plateau and its surroundings), and regions where it is difficult to set up sounding stations (e.g., vast uninhabited regions of the Tibetan Plateau and the oceans). Once the nationwide

ADDRS network is established, it will provide extensive atmospheric vertical profile observations for regions where such data are currently scarce, thereby supporting the NWP and atmospheric science research. This advancement will be of great significance for improving the understanding of weather patterns in China and enhancing the accuracy of the NWP.

[Figure]

**Fig. R3 Spatial distribution of the landing points (colored dots) for sounding stations (black triangles) across China from February 2022 to January 2024. Different colors represent different months.**

*(3) There is little information about availability of these data to the public. They are potentially valuable for weather centers around the globe or for atmospheric climate reanalysis. It would be important for the reader to know whether these data can be accessed and used under a general public license.*

**Authors: -** The field experiments data of ADDRS involves data policies and can only be shared to a limited extent. It can be shared on a small scale through cooperation between both parties. The CMA is actively studying and formulating the sharing strategy. The authors will actively strive to promote the sharing of this data.

*3. Minor Comments:*

*1) Table 3: Please explain what "Airspeed" means. Is it the data transmission rate in bits per second?*

**Authors: -** Yes, Thank you for your rigorous review. There is a word error. The "Airspeed" meaning is not clear, we will replace the "Data Transmission Rate" On Page 9, Table3 in the revised manuscript.

*2) RC1:L295: It is mentioned that the effect of radiation in the drift phase is greater*

*than during ascent/descent. Can you quantify that, ideally with a plot or table showing the increased measurement uncertainty during the drift phase. It is unclear to the reader how the CFD correction model works and how it performs.*

**Authors: -** Thank you careful reading and constructive suggestions. Considering that this manuscript is primarily dedicated to the introduction of the "ADDRS" entire system, which mainly encompasses instruments and hardware system like radiosondes, dual-mode balloon, drifting controller, and so on, field experiments, data quality, data applications, etc. Regarding Section 3.4, "ADD" measurement technology, it is designed to mitigate temperature errors during the drift phase, which goes beyond the traditional ascent - phase sounding, and to effectively detect meteorological elements such as wind during the descending phase. Consequently, we intend to comprehensively showcase the outcomes of this part in a separate SCI paper. As a result, the Table.R1 and Fig.R4,R5,R6,R7 will not be presented in the revised manuscript. We sincerely hope for your kindly understanding.

We draw a Fig. R4,It shows how the CFD correction model works

We draw a Fig.R5, It shows how the the CFD correction model performs, during one observation process of ADDRS, the temperature of raw data(red line), It is obviously can see the effect of radiation in the drift phase is greater than during ascent/descent during the one process of ADDRS. the temperature profile by CFD correction model decreased effect of radiation random error(blue line).

[Figure]

**Fig. R4 the operating mode of CFD correction model .**

[Figure]

**Fig. R5 decreased effect of radiation error during one observation process of ADDRS.**

*RC3:L317-318 "Comparative tests led to the selection of conical parachutes" Please provide more details, such a a diagram, dimensions and any events.*

**Authors: -**

More than ten comparative tests were carried out for the improvement of parachute materials, ventilation structures, main parachute dimensions, etc. The Table.R1 presents two typical groups of tests. After statistics, the oscillation angled of conventional parachute is greater than 2°. The oscillation angle of conical parachute is less than 1°, which can ensure the rationality of the Gaussian filtering correction filter window for the horizontal wind in the descent section. Meanwhile, a larger main parachute size can reduce the descent speed. Therefore, without considering the cost, "Large conical parachute" is recommended for the descent section detection of ADDRS.

**Table.R1 Comparative tests of the selection of conical parachutes**

| Comparative tests | Parachute Type | Oscillation Angle (°) | Oscillation Velocity V (m/s) |
|---|---|---|---|
| Test 1 | Conventional parachute | 3.9 | 1.6 |
| | Conical parachute | 0.8 | 0.5 |
| Test 2 | Conventional parachute | 2.5 | 1.3 |
| | Large conical parachute | 1.8 | 0.5 |
| | Small conical parachute | 0.3 | 0.15 |

[Figure]

**Fig.R6 Physical images of conventional parachute, small conical parachute, large conical parachute.**

*2)319 "As a result, RDSS achieves a stable descent speed of approximately 6 m/s ± 1 m/s" Is this true even in the stratosphere? Please show some example profiles and/or mean and standard deviation profiles. Ingleby et al (2022) report much faster descent speeds in the stratosphere (even with a parachute) and a lot of variation from one flight to another.*

**Authors:** -Thank you for your meticulous and rigorous guidance. on Page 12, L318 in revised manuscript. Ingleby et al (2022) is cited. As the Fig.R6 shows by using both conventional parachutes and conical parachutes simultaneously, a stable descent speed of 6 m/s±1 m/s can be achieved in UTLS (upper troposphere and lower stratosphere (7-17km)).

[Figure]

**Fig.R7 the ascent speed curve of conventional parachute、small conical parachute、large conical parachute**

*3)RC1:Table 4.2: The assessment is rather crude, particularly for humidity. Having just one value for the whole troposphere is insufficient by today's standard. Can you give a profile of the humidity measurement uncertainty?*

**Authors: -** Thank you for your constructive suggestions. We have another manuscript is under review:

Beidou Navigation Radiosonde Observation Experiment and Data Evaluation, Lebao Yao, Dan Shen, Xin Sun, Donghai Wang, Xiaozhong Cao, Jincheng Wang, Dan Wang, Chunyan Zhang, Qiyun Guo. Submitted to Atmospheric Research

which is assessment "profile of the humidity measurement uncertainty"

*RC3:IMPORTANT: Please provide mean differences for temperature. What quality control (including comparison with ERA5) was used to exclude outliers? What 'saturation vapour pressure' equation is used for RH? As discussed with Lebao Yao, I think there are differences in the definition of RH between the radiosonde and ERA5. It is cleaner to start from ERA5 specific humidity and temperature and then make sure*

*you are comparing like with like. Note. The 'drift' temperature differences seem too large to be usable (despite the processing described above). It would be useful to know the mean wind speeds for the different phases.*

**Authors: -** Thank you for your constructive and insightful feedback. On Page 15-16, L382 of the revised manuscript, we have provided the bias and standard deviation of the U and V wind components, temperature, and relative humidity, further distinguishing between daytime and nighttime observations.

Regarding quality control, we have applied an outlier rejection step based on the temporal variability of the observation data itself. In addition, in Yao's analysis, a bi-weight algorithm was further applied to the O – A samples by comparison with ERA5 reanalysis, which allowed additional outliers to be excluded.

For relative humidity, the ADDRS radiosonde uses the Goff (1957) SVP equation, while ERA5 employs the Buck SVP equation. The choice of different saturation vapour pressure equations can lead to substantial differences under low-temperature conditions. We have discussed this issue in depth with Yao, and our results are now consistent with his most recent recalculation. Specifically, Yao recalculated ERA5 relative humidity using ERA5 temperature and specific humidity with the Goff (1957) SVP equation. In his first recalculation, however, Yao followed the IFS documentation, in which relative humidity is recalculated using the SVP equation over water and over ice in different temperature ranges. As a result, the recalculated ERA5 RH values produced only negligible changes in the O-A bias and standard deviation. We later realized that radiosonde humidity sensors typically use the SVP equation over liquid water only, even under very low-temperature conditions. Accordingly, Yao performed a second recalculation using the SVP over liquid water exclusively, thereby ensuring a more consistent comparison.

We greatly appreciate your insightful observation. After this adjustment, the O-A bias and RMSE of relative humidity under low-temperature conditions are significantly reduced and become much more physically reasonable.

And the 'drift' temperature differences is definitely too large to be usable especially during the day. The CFD correction model is a preliminary attempt. After deviation correction, the temperature uncertainty of the drift section decreased to 3.09K. The relevant technical achievements are being sorted out. However, as the operational Beidou radiosonde involves six models of GTH, we are combining it with the temperature correction method of CF-06-AH, hoping to develop some new temperature correction methods for the drift section.

As of now, the results of the horizontal wind still need to be further studied in accordance with the assessment method of UAⅡ 2022.

**10.1.9.2 CF-06-AH**

The CF-06-AH sonde has one of the smallest temperature measurement biases of all the sondes involved in the UAII 2022, with a mean measurement error of $|\bar{\delta}|<0.05\,\mathrm{K}$, particularly for nighttime flights (see Figures 11.1 and L.16). It is this characteristic that lets it be the only sonde found to be "fit-for-purpose" with respect to the (most challenging) ORUC Goal value associated with the application area "Atmospheric Climate Forecasting and Monitoring", albeit still only in the UTLS for nighttime flights (see Table 11.4).

The CF-06-AH sonde shows a wet measurement bias of $\bar{\delta}\approx+3\,\%\mathrm{RH}$ up to the tropopause. For several flights where the sonde passed through a 100 %RH layer (e.g. F26, F69, F76), this sonde reports elevated relative humidity readings throughout the stratosphere (up to 15 %RH for certain flights, see Figure L.17).

174

**Fig.R7 the result of CF-06-AH in UAⅡ 2022**

**Table 5. Comparative analysis of after-quality control of ADDRS radiosonde data and ERA5.**

| | | | U (m/s) | | V (m/s) | | T (K) | | RH (% RH) | |
|---|---|---|---|---|---|---|---|---|---|---|
| | | | bias | std | bias | std | bias | std | bias | std |
| Ascent | Day | Above tropopause | 0.03 | 1.73 | 0.11 | 1.91 | −0.4 | 1.04 | / | / |
| | | Below tropopause | 0.13 | 1.41 | 0.08 | 1.47 | 0.04 | 0.69 | 1.32 | 8.57 |
| | Night | Above tropopause | −0.02 | 1.72 | −0.01 | 1.87 | 0.01 | 1.02 | / | / |
| | | Below tropopause | 0.13 | 1.4 | 0.02 | 1.46 | 0.08 | 0.66 | 1.62 | 8.32 |
| Drift | | above tropopause | / | 3.32 | / | 3.22 | / | 3.09 | / | / |
| Descent | Day | Above tropopause | 0.16 | 1.74 | −0.02 | 1.9 | 0.9 | 1.14 | / | / |
| | | Below tropopause | 0.14 | 1.65 | 0.01 | 1.7 | 0.42 | 0.75 | −1.7 | 10.12 |
| | Night | Above tropopause | 0.16 | 1.73 | 0.03 | 1.84 | 0.21 | 1.06 | / | / |
| | | Below tropopause | 0.15 | 1.67 | 0.05 | 1.67 | 0.04 | 0.73 | 0.5 | 10.47 |

*4)L368: track the occurrence of the entire convective system and the changes .. in real time... is a bit strong wording, given the fine grained structure visible in the RADAR picture which is by no means resolvable with a few radiosondes, even with drifting capability.*

**Authors: -** We have been revised 5.1 The applications in weather analysis relevant

expressions"The drifting trajectories of ADDRS(Beidou) radiosondes were from west to east, which aligned with the movement and development direction of severe convection. The descending radiosondes of ADDRS provides effective monitoring and significant insights in track the occurrence of the convective system and the changes in the ambient field. " on Page 16, L389-392 in revised manuscript.

*5)L425: results from one month are not conclusive. There should be at least three months (e.g. used as minimum by ECMWF when upgrading their system), ideally from different seasons, to cover any dependency on the annual cycle.*

**Authors: -** Sorry, At present, the assimilation experiment has only been completed for one month, which indeed cannot fully demonstrate its effect. Combining the optimization of observation errors and sparsity schemes, we are currently studying a comprehensive optimization plan. We plan to optimize the assimilation parameters such as observation error and sparsity before conducting a one-year batch trial.The relevant results may be compiled into another manuscript. We sincerely hope to invite you to offer insightful comments.

*6)L534: The reduction of the forecast trajectory error in one typhoon event must be considered anecdotal. While encouraging it is certainly not science-grade evidence of an improvement.*

**Authors: -** Thank you for your constructive suggestions. The science-grade evidence of an improvement is a long process. But we will make persistent efforts to prove it. And we cite relevant paper on Page 27, L749-752 in revised manuscript..

**REFERENCES:**

Wen, Q. S., Zhang, X. F., Hu, S., Zhao, P. T., Zhong, S. X., Liu, Z. Y., Zhao, Z. K., Liang, J. H., Dai, G. F., Zhang, C. Z., Li, M. J., Huang, L.: Collaborative assimilation experiment of Beidou navigation radiosonde and drone-dropped radiosonde based on CMA-TRAMS, Atmos. Oceanic Sci. Lett., 18(2), https://doi.org/10.1016/j.aosl.2024.100555, 2025.

*7)L550: It is not obvious from the material presented that the RDSS really reduces costs compared to conventional radiosondes. This needs more detailed explanation.*

**Authors: -** In fact, one ADDRS can provide 'Ascent-Drift-Descent' radiosonde data in which executed through one balloon launch. The cost is about equal to operational radiosonde observations(Vaisala RS41+Totex 600g balloon). However, the application value of its data is still under continuous exploration. The relevant description has been modified as follows" ADDRS created a new model, which is achieved a three-phase sounding at one launch by effectively incorporating ascent profile sounding, stratospheric drift and descent soundings." on Page 22, L564-566 in revised manuscript.

*8)L567: This statement on data availability is unacceptable by today's open science standards.*

**Authors: -** The field experiments data of ADDRS involves data policies and can only be shared to a limited extent. It can be shared on a small scale through cooperation between both parties. And we modify "L567 Requests for data that support the findings of this study can be sent to luohw_1@163.com."to "The CMA is actively studying and formulating the sharing strategy. The authors will actively strive to promote the sharing of this data. According to the relevant policies of the CMA, this data currently cannot be widely shared. For data sharing, you can contact us to discuss cooperation and data sharing." on Page 23, L594-597 in revised manuscript.

**Response to Referee #2's comments**

*Referee #2: egusphere-2025-2012*

*Synopsis: The Round-trip Drifting Sounding System (RDSS) proposed in this study demonstrates significant innovation and holds promising application potential. It is recommended for publication after minor revision. The RDSS's key advantage lies in completing a continuous atmospheric observation cycle of "ascent (1 hour) - drifting (4 hours) - descent (1 hour)" with a single balloon launch. Notably, the drifting phase provides continuous observation of the lower stratosphere, while the descent phase provides atmospheric vertical profiles over remote areas far from the launch site. This effectively addresses the gap in traditional sounding observations at 06 and 18 UTC. Furthermore, by utilizing long-distance drifting, RDSS expands the capability for vertical atmospheric sounding in remote regions through the descent observations. Field experiments conducted in the Yangtze River Basin successfully tackled several key technical challenges. Compared to China's previous-generation L-band sounding*

*system, RDSS exhibits reduced observation errors, particularly in tropospheric wind fields and humidity. Additionally, RDSS possesses a degree of mobility, suggesting potential for targeted observation campaigns. The study also presents preliminary results from RDSS data assimilation and forecast impact experiments, indicating a significant improvement in the skill of 12-24 hour accumulated precipitation forecasts initialized at 06UTC and 18UTC.*

**Authors: -** We thank Referee #2 (RC2) for her/his positive evaluation of our research. We have addressed each of reviewer's comments and inquires point by point and revised the manuscript carefully. And We sincerely thank the Referee #2 (RC2) for her/his valuable feedback that we used to improve the quality of the manuscript. Based on the comments and suggestions from the RC2, we have carefully and substantially revised the manuscript. We have included a detailed response addressing each of comments. The comments are laid out below in italicized font and specific concerns have been numbered. Our response is given in normal font and changes/additions to the manuscript are given in the blue text. The line numbers correspond to the revised manuscript without the track changes displayed. For precise details on the modifications made in the latest version, please consult the supplementary material, where track changes are enabled.

*Specific Recommendations for Revision:*

1. *Table 1: The horizontal lines in the three-line table vary in thickness. It is recommended to standardize the line weight.*

**Authors: -** We thank the RC2 for the valuable and creative comments. Your statement is correct, we standardize the line weight of Table 1 on Page 4, L105 in revised manuscript.

**Table 1. Main instruments and key functions of ADDRS.**

| No | Subsystem | Instruments | Key Function |
|---|---|---|---|
| 1 | "ADD" subsystem | dual-mode balloon | "outer balloon" as ascent carrier, "inner balloon" as drift carrier |
| 2 | | parachute | parachute as the carrier of the descent phase |
| 3 | | drifting controller | Adaptive control of drift and descent |
| 4 | | radiosonde | The temperature, pressure, humidity, wind measurement meet the demand for long-term stratospheric observation |
| 5 | Ground operation control subsystem | Ground station | ground inspection ground check, balloon inflation, launch, and other tasks before the equipment is launched |
| 6 | | Multichannel sounding receiver | 8 channels receive radiosonde data simultaneously |
| 7 | | control command transmitter | In the weather-sensitive area without a station, the active fusing drifting controller is carried out and the descent measurement is started |
| 8 | | operational management system | Real-time acquisition, transmission, quality control, and timely delivery of control instructions for ADDRS data, providing real-time high-quality data to weather analysis and numerical prediction models |

2. *Table 4: Words are hyphenated across lines. It is recommended to adjust the table format to prevent word breaks.*

**Authors: -** We thank the RC2 for the valuable and creative comments. Your statement is correct, we prevent word breaks,the new Table 4 on Page 9, L253 in revised manuscript.

**Table 4. The evaluation results of GTH3 radiosonde temperature, pressure, relative humidity, wind and geopotential height in WMO Instruments and Observation Methods Report No. 143. page.150(Note: The data are in the form of $\Lambda_{c,L}{}^{\delta_{c,L}}_{\sigma(\delta)} \pm \epsilon_{c,L}$ , where $\Lambda_{c,L}$ represents the individual measurement root mean square error, $\epsilon_{c,L}$ denotes the measurement uncertainty, $\delta_{c,L}$ is the measurement error, and $\sigma(\delta)$ indicates the measurement standard deviation. The planetary boundary layer (PBL) ranges from surface to 2 kilometers; the free troposphere (FT) ranges from 2 kilometers to the tropopause 12 kilometers are in the ; the upper troposphere/lower stratosphere (UTLS) ranges from 7 kilometers to 17 kilometers; the middle and upper stratosphere (MUS) is above 17 kilometers up to the bursting point of the sounding balloon. )**

| Time | Height | Atmospheric temperature [K] | Relative humidity [%RH] | Geopotential height [m] | Pressure [hPa] | Wind (horizontal)direction [°] | Wind (horizontal)speed [ms⁻¹] | Wind (horizontal)vector [ms⁻¹] |
|---|---|---|---|---|---|---|---|---|
| Day | PBL | $0.18^{-0.05}_{0.17}\pm0.03$ | $7.00^{-5.43}_{4.41}\pm0.74$ | X | X | X | X | X |
| | FT | $0.12^{+0.05}_{0.11}\pm0.04$ | $8.75^{-3.50}_{8.02}\pm0.60$ | $5.9^{+2.0}_{5.5}\pm1.8$ | $0.4^{-0.0}_{0.4}\pm0.1$ | $3.6^{-0.4}_{3.6}\pm0.2$ | $0.2^{-0.0}_{0.2}\pm0.0$ | $0.3^{+0.2}_{0.1}\pm0.0$ |
| | UTLS | $0.09^{+0.01}_{0.08}\pm0.03$ | $7.73^{-1.55}_{7.58}\pm0.40$ | $13.2^{+10.0}_{8.6}\pm3.8$ | $0.4^{-0.3}_{0.2}\pm0.1$ | $2.5^{-0.2}_{2.5}\pm0.3$ | $0.2^{-0.0}_{0.2}\pm0.0$ | $0.3^{+0.2}_{0.2}\pm0.0$ |
| | MUS | $0.27^{-0.22}_{0.16}\pm0.10$ | $1.69^{+1.48}_{0.82}\pm0.46$ | $29.5^{+23.4}_{17.9}\pm4.2$ | $0.3^{-0.2}_{0.1}\pm0.0$ | $6.1^{-0.4}_{6.1}\pm0.2$ | $1.3^{-0.0}_{1.3}\pm0.0$ | $1.5^{+0.3}_{1.5}\pm0.0$ |
| Night | PBL | $0.38^{+0.18}_{0.34}\pm0.05$ | $4.72^{+0.74}_{4.66}\pm0.15$ | X | X | X | X | X |
| | FT | $0.15^{+0.02}_{0.15}\pm0.02$ | $6.41^{+2.16}_{6.03}\pm0.11$ | $5.8^{+0.4}_{5.8}\pm0.4$ | $0.5^{+0.1}_{0.5}\pm0.2$ | $2.6^{-0.2}_{2.6}\pm0.2$ | $0.2^{-0.0}_{0.2}\pm0.0$ | $0.2^{+0.2}_{0.1}\pm0.0$ |
| | UTLS | $0.12^{+0.06}_{0.10}\pm0.05$ | $6.82^{+3.70}_{5.74}\pm0.26$ | $11.5^{+7.7}_{8.6}\pm3.4$ | $0.3^{-0.1}_{0.2}\pm0.1$ | $2.4^{-0.1}_{2.4}\pm0.1$ | $0.2^{+0.0}_{0.2}\pm0.0$ | $0.2^{+0.2}_{0.1}\pm0.0$ |
| | MUS | $0.10^{-0.03}_{0.10}\pm0.02$ | $1.71^{+1.54}_{0.74}\pm0.28$ | $26.7^{+20.7}_{16.8}\pm4.2$ | $0.1^{-0.1}_{0.1}\pm0.0$ | $4.5^{-0.6}_{4.4}\pm0.2$ | $0.2^{-0.0}_{0.2}\pm0.0$ | $0.4^{+0.3}_{0.3}\pm0.0$ |

3. ***Data Assimilation Experiments:*** *The assimilation application experiments using RDSS data in this paper are relatively limited. It is recommended that subsequent research focuses on: (a) Further refining assimilation techniques for RDSS second-level data (e.g., data thinning methods); (b) Conducting more extensive numerical forecast assimilation impact experiments utilizing additional observational data; (c) Conducting a thorough evaluation of the forecast skill improvement offered by RDSS compared to China's conventional L-band sounding system.*

**Authors: -**Thank you for the suggestions of the RC2. In the subsequent research, we will evaluate and improve the data on numerical forecasting for the ADDRS about your suggestion.

4. *"Zhuang Z R, Wang R C (2019), Wang J C, et al." - Literature duplication.*

**Authors: -**Thank you for the suggestions of the RC2. We apologize for the inconvenience caused to you. The revised manuscript's REFEERNECE has been reorganized.

5. *The two attached papers also studied and confirmed the impact and value of the RDSS data on numerical forecasting. It is recommended to include them.*

*ZHANG Xin, WANG Qiuping, MA Xulin, et al. 2025. The Influence of New Round-Trip Drifting Sounding Observation on the Quality of Numerical Prediction in the Middle and Lower Reaches of the Yangtze River [J]. Chinese Journal of Atmospheric Sciences, 49(1): 245−256. doi:10.3878/j.issn.1006-9895.2304.22224*

*Zhang, X., Sun, L., Ma, X., Guo, H., Gong, Z., Yan, X. Can the Assimilation of the Ascending and Descending Sections' Data from Round-Trip Drifting Soundings Improve the Forecasting of Rainstorms in Eastern China? Atmosphere 2023, 14, 1127. https://doi.org/10.3390/ atmos14071127*

**Authors: -** We read the two papers. It was of great help to us. In the subsequent research, And we will evaluate and improve the data on numerical forecasting for the application of ADDRS.

**We will add the two paper attached in REFERENCES on Page 28,29, L799-804:**

ZHANG Xin, WANG Qiuping, MA Xulin, et al. 2025. The Influence of New Round-Trip Drifting Sounding Observation on the Quality of Numerical Prediction in the Middle and Lower Reaches of the Yangtze River [J]. Chinese Journal of Atmospheric Sciences, 49(1): 245−256. doi:10.3878/j.issn.1006-9895.2304.22224

Zhang, X., Sun, L., Ma, X., Guo, H., Gong, Z., Yan, X. Can the Assimilation of the Ascending and Descending Sections' Data from Round-Trip Drifting Soundings Improve the Forecasting of Rainstorms in Eastern China? Atmosphere 2023, 14, 1127. https://doi.org/10.3390/ atmos14071127

**Response to Anonymous Referee #3**

*Review of Development and application of the Round-trip Drifting Sounding System (RDSS) by Cao et al*

*1.  **General comment:** , I have just completed a rereview of "The Beidou Navigation Radiosonde Observation Experiment and Data Evaluation" by Lebao Yao et al (with Xiaozhong Cao as a co-author) submitted to Atmospheric Research. When the request for review from Copernicus arrived I clicked accept thinking that this was the other paper. The two papers do not seem to have been coordinated as I would expect: in terms of content - there is some overlap, the name of the sounding system differs and there is no reference to Yao et al in this manuscript. I find the 'Round-trip' in the title a misleading name. In English a round-trip finishes at the point of origin - not the case with the RDSS. Yao et al use the term Beidou Navigation Radiosonde. In places this manuscript refers to the 'ADD' (Ascent-Drift-Descent) system instead of RDSS.*

*ADD seems a more accurate name to me. More explanation of the overall design would be useful. What is the purpose of the 'drift' stage? What determines how long it is?*

**Authors: -** Yes, as a co-author of "The Beidou Navigation Radiosonde Observation Experiment and Data Evaluation", I am acquainted with the work "The Beidou Navigation Radiosonde Observation Experiment and Data Evaluation" by Lebao Yao et al. In this manuscript, the data support for the comparison results between 4.2 Data quality evaluation and ERA5 was primarily provided by Yao ,et al.

We sincerely appreciate the reviewers' comments, which have put forward excellent suggestions for standardizing the name of our system. Initially, we interpreted the Chinese term "round - trip" as the process of returning from the sky to the ground. In numerous Chinese-language papers with publicly available data and applications, this understanding did not lead to any ambiguity. For instance, several papers, including (Zhang, X et al., 2025; Zhang, X. P et al. 2023), directly translated our evolving system from Chinese into "Round - Trip Drifting Sounding". Additionally, (He, Y et al. 2024) referred to our system as "Round - Trip Intelligent Sounding System, RTISS".

However, we did not recognize that in an English context, a "round-trip" concludes at the starting point, which is not applicable to the RDSS. This oversight caused a semantic discrepancy between Chinese and English interpretations. Combined with your advice and descriptions in Yao,et al and comprehensively considered by the ADDRS research team, We realize to the drifting phase observation represents just one aspect of the system's application and does not warrant special emphasis. The three - stage detection mode of ascent, drift, and descent is a distinctive feature of our system, And "Ascent-Drift-Descent Radiosonde System" (ADDRS) is more conducive to accurate comprehension. So, we have decided to modify the system's name from "Round-Trip Drifting Sounding System (RDSS)" to "Ascent-Drift-Descent Radiosonde System (ADDRS)".

*2. How much does it add to the batteries required? Will the drift and descent stages be reported on the GTS? Is so what format will be used for the drift stage? At recent WMO meetings (eg TECO 2022) there has been discussion of reducing the environmental impact of radiosondes and other instruments. The reduced weight of the new radiosonde is a step in that direction, but has there been any effort to reduce plastic use or the toxic elements in circuit boards? Are there plans to refurbish and reuse some of the radiosonde instruments?*

**Authors: -** Yes, similar issues have been coordinated with Yao et al. The CMA observation stations in Qingyuan and Shantou, Guangdong, and Xilinhot, Inner Mongolia, have been exchanging data from their Beidou Navigation radiosondes with

the World Meteorological Organization (WMO) via the Global Telecommunication System (GTS) in BUFR format since 00 UTC on January 1, 2024 and January 1, 2025, respectively. Currently, the data format agreement for the free-drift and descent segments is still under internal review by CMA. We hope to obtain valuable opinions from relevant technical groups such as the WMO UT-UAM Technical Group (with Guo Qiyun as a member) and the INFCOM Information Transmission Study Group, and strive to incorporate the relevant content into the WMO GTS data exchange policy.

In terms of reducing the environmental impact of radiosondes and other instruments, we have redrawn Table 3 on Page 9, L247 of the revised manuscript. This table shows that the China Meteorological Administration (CMA) has significantly reduced material usage by upgrading the operational radiosondes from GTS1 to GTH3. During a single sounding process, the weight of lithium thionyl chloride batteries/lithium manganese batteries has decreased by approximately 210 grams (single profile) or 160 grams ("ADD" three-phase profile), the EPS (expanded polystyrene) packaging foam has decreased by about 40 grams, and the metal circuit board has decreased by about 30 grams. Based on the current 131 radiosonde observation stations of CMA, with two regular observations per year ($2 \times 365$ days) and additional observations during special weather conditions, approximately 100,000 Beidou navigation radiosondes are used throughout the year, equivalent to consuming 16 to 21 tons of lithium thionylchloride batteries/lithium manganese batteries materials, 4 tons of packaging foam, and 3 tons of metal circuit boards.

We have made efforts to reduce the use of plastic and the presence of toxic elements in circuit boards. We have carefully reviewed the test report provided by the manufacturer for the GTH3 radiosonde circuit board, and the results show that the contents of lead (Pb), cadmium (Cd), mercury (Hg), hexavalent chromium (Cr(VI)), polybrominated biphenyls (PBBs), polybrominated diphenyl ethers (PBDEs), and phthalates (DBP, BBP, DEHP, DIBP) all meet the limit requirements of the RoHS Directive 2011/65/EU and its revised Directive (EU) 2015/863. At the same time, we have also conducted harmful substance tests on the lithium manganese batteries used in the GTH3 radiosonde. The test items include lead (Pb), mercury (Hg), cadmium (Cd), hexavalent chromium (Cr(VI)), polybrominated biphenyls (PBB), polybrominated diphenyl ethers (PBDE), di(2-ethylhexyl) phthalate (DEHP), butyl benzyl phthalate (BBP), dibutyl phthalate (DBP), and diisobutyl phthalate (DIBP). The test methods are based on IEC 62321-4:2013 + amd:2017, IEC 62321-5:2013, IEC 62321-7:2017, IEC 62321-6:2015, and IEC 62321-8:2017 standards, using ICP-OES/AAS, UV-Vis, and GC-MS analytical techniques. The test results were all undetectable (<MDL).

In addition, the China Meteorological Administration (CMA) plans to refurbish and reuse some radiosondes. Currently, the packaging material for the Beidou navigation radiosonde is expandable polystyrene (EPS), but the CMA plans to replace EPS with degradable materials such as all-starch foam boards, and will conduct relevant

transportation and vibration strength tests on the packaging structure. Regarding the recycling mechanism, we are exploring the formulation of new operation policies. For example, based on the landing location of the radiosonde, if it is in an inhabited area, the local environmental protection department can be coordinated to organize relevant personnel for recycling, and the equipment can be sent back to the manufacturer at a lower cost for refurbishment and reuse. The circuit boards, packaging foam, and batteries after recycling can all be reused, but the recalibration of temperature, humidity, and pressure sensors is an important technical task that requires further research and standardization.

The GTH3 radiosonde uses the HC103M2 capacitive thin-film polymer humidity sensor from the Austrian E+E manufacturer, and GTH4 is same as CF-06-AH uses the P14 capacitive thin-film polymer humidity sensor from the Switzerland IST manufacturer. Now, CMA have been developed a heating and measure temperature type humidity sensor similar to Vaisala RS41 has undergone over six hundred comparison tests, and the performance results have been stable. It is expected that the humidity sensor of the radiosonde will be upgraded by refurbish in 2026-2027.

**Table 3. Comparison of parameters among RS41, GTH3 and GTS1 radiosonde.**

| Radiosonde type | Positioning method | Volume (mm³) | Weight (g) | Transmitting power (mW) | Working time (min) | Data Transmission Rate(bps) | Battery Weight (g) | Foam Packaging Weight (g) | Circuit boards Weight (g) | Reduce toxic elements in circuit boards |
|---|---|---|---|---|---|---|---|---|---|---|
| GTS1 radiosonde | Radar positioning | 190×90×245 | <400 | 400≤ | >120 | 1200 | <250 | <70 | <80 | / |
| GTH3 | Beidou Equinox I | 155×65×60 | <120(for one profile) | 100≤ | >240 | Optional,2400,4800,9600 | <40 | <30 | <50 | Test Lead (Pb), Cadmium (Cd), Mercury (Hg), Hexavalent Chromium (Cr(VI)), Polybrominated Biphenyls (PBBs), Polybrominated Diphenyl Ethers (PBDEs), Phthalates (DBP, BBP, DEHP, DIBP), and the results comply with the limits set by RoHS Directive 2011/65/EU with amendment (EU) 2015/863. |
| | | 155×65×60 | <170(for "ADD" three phase profile) | 100≤ | >640 | | <90 | <30 | <50 | |
| Vaisala RS41 | GNSS u-blox G7020 | 155×60×46 | 109 | 60 | >240 | 4800 | 76 | / | / | / |

*3.There should also mention of the WMO Global Basic Observing Network (GBON, https://community.wmo.int/en/activity-areas/wigos/gbon ) as CMA is planning to use these drift radiosondes operationally*

*GBON technical regulations specify that there should be soundings to 30 hPa and a subset to 10 hPa.*

*To what extent would this be met in the tests so far?*

**Authors: -** Thank you for your constructive and insightful feedback. The ADDRS not only as an important part in upper-air operational sounding in CMA, but also will

become a part of WMO Global Basic Observing Network (GBON, https://community.wmo.int/en/activity-areas/wigos/gbon ) is cited in L86 and L758, As the Fig.R7, The CMA plans to jointly carry out the ADDRS with Hong Kong, Macao, Pakistan and Mongolia RAⅡMembers from 2025 to 2027 to conduct targeted observations of typhoons and other severe weather, responding to the WMO's Early Warnings for All.(EW4All) and    make contributions to GBON data.

| RA II-18-I-DP-2 | LTG-2/SO 2.3 | Cg-19 Resolution 20 (Cg-19) | 8 | An emerging and cost-effective technology for upper air measurements – | (1) Promote the application of RDSS observations in typical disaster | (1) Number of RA II Members accessing adequate RDSS data | (2025-2027) (1) Promote RDSS data application among RA II Members (2025-2027) | RDSS-hosting Members |

| | | | | Round-trip Drifting Sounding System (RDSS) | weather such as typhoons and severe convection (2) Evaluate the benefit of RDSS application in operational sounding observations and NWP cases (3) RDSS data used by RA II Members | (2) Number of RDSS data applied in typhoons and severe convective and other typical disaster weather | (2) Collect and sort out detailed typhoons and severe convective weather NWP cases (2025-2027) (3) Support RA II Members to perform RDSS observations (including mobile stations) (2026-2027) (4) Conduct meeting(s) to invite RA II Members to share experiences in the implementation of RDSS observations and NWP cases (2025-2027) | |

Fig.R8 2025-2027 Operating Plan of resolution on the WMO RAII-18

30hPa corresponds approximately to an altitude of around 24km, and 10hPa corresponds approximately to an altitude of around 30km.

Revised 4.1    Field experiment in the middle and lower reaches of the Yangtze River Region on Page 14, L351-354 "From March to September 2021, 2,427" add "radiosonde launches were conducted with 2,281 classified as effective launches,of which 1,772 achieved successful drifting. Among these 1587 launches resulted in drifting for more than 4 hours. The overall drifting success rate was 77.68% with a 4-hour drifting success rate of 69.57%. " The 1,772 successful drifts were analyzed, and 937 were successful during the daytime, accounting for 52.88%. The successful nighttime drifting was 835 times, accounting for 47.12%, and the effect of daytime was better than nighttime.

The drift heights ranging from < 75hPa(<18km) to 15hPa-10hPa(28km-32km) were statistically analyzed. The proportion of outer ball explosion heights within 15hPa-10hPa(28km-32km) was the largest (42.89%). Secondly, the proportion of 20hPa-15hPa(26km-28km) is 40.35%, the proportion of 30hPa-20hPa(24km-26km) is 8.97%, and the total proportion of beyond 30hPa is 92.21%. Therefore, the drift height should basically meet the 30hPa requirement of GBON.

However, drift ability of beyond 10hPa(31km) is still a difficulty. In response to M. Fujiwara et al., 2025: Justification for high-ascent attainment, experiments with stratospheric balloons (3000g weather balloons) 2-3 times a week have been carried out on the basis of 1600g weather balloons for the elevation of GRUAN station Xilin hot station in China. The CMA will focus on breaking through the drift technology above 10hPa(31km) in the next step.

[Figure]

**Fig 6. The sounding-forecasting interactive network experiment (2021): (a) drifting height; (b) descent height.**

*3. One could read this manuscript and think that NWP revolves almost entirely around radiosonde profiles - not true. There is one mention of aircraft reports and almost nothing about satellite observations. In practice satellite observations are now hugely important and radiosondes are less so although still very useful (partly for validation/verification). Below are three references (of many) to support this view.*

*Bauer, P., Thorpe, A. & Brunet, G. The quiet revolution of numerical weather prediction. Nature 525, 47-55 (2015). https://doi.org/10.1038/nature14956*

*Bormann N, Lawrence H, Farnan J (2019) Global observing system experiments in the ECMWF assimilation system. ECMWF Technical Memorandum 839, 24 pp. https://doi.org/10.21957/sr184iyz*

*https://community.wmo.int/en/meetings/8th-wmo-impact-workshop-home (2024, final report and presentations)*

**Authors: -** Yes, the significance of satellite observation is unquestionably indubitably. The above three references have been cited in L60-61 of the revised draft: "the radiosonde observations are still very useful for validation/verification satellite observations(Bauer et al,2015; Bormann et al,2019;WMO et al,2024). " Meanwhile, we have also carried out some cross-testing work between satellites and radiosondes. on Page 14, L361-L364 of the revised draft, "the balloons drifting of ADDRS

radiosondes data in the above tropopause has been very useful for verification FY-3D satellite temperature and humidity profiles(Zhou.x.et al., 2023; Zhou.x.et al., 2024) can be referred to.

***Detailed comments***

*1)17 "Meteorological sounding primarily refers to the balloon-borne radiosonde"*

*The satellite community might disagee. Rewrite.*

**Authors: -**Thank you for your suggestions. We agree with the reviewer's comment.

The sentence "Meteorological sounding primarily refers to the balloon-borne radiosonde" in preprint manuscript has been modify to "The balloon-borne radiosonde sounding constitutes a crucial component of meteorological sounding"on Page 2, L19-20 of revised manuscript.

*2)21 "zero-pressure dual-mode meteorological balloon" what does zero-pressure mean?*

**Authors**: - Thank you for your suggestions. The term "zero-pressure" in L23,Table 1,,L109-L110 of the preprint manuscript has been removed. Uniformly referred to as "dual-mode balloon". The following reference to zero-pressure balloons is sourced from https://www.nasa.gov/scientificballoons/types-of-balloons/: "Zero-Pressure Balloons: These balloons are open at the bottom and have open ducts hanging from the sides to allow gas to escape, preventing pressure buildup inside the balloon as gas expands during ascent above Earth "s surface. The duration of this type of balloon is limited due to gas loss, primarily caused by the balloon "s day/night cycling."

*3)18 "uppe-rair" > "upper-air"*

**Authors: -** Thank you for your rigorous review. This is a spelling word mistake. in L18 of the preprint manuscript "uppe-rair"  has been changed on Page 2, L24" upper-air ", and all spelling word mistakes in the revised manuscript have been proofread.

*4)46-47 "Since the 19th century, the balloon-borne radiosonde has served as the primary tool for direct measurements of upper-air meteorological elements below 30 km and is extensively utilized globally (Gallice et al., 2011)." The use of radio with balloon soundings only started in the 1930s or so (20th century), with early operational systems in the 1940s. A better reference would be Pettifer (2009). Pettifer R (2009) From observations to forecasts, part 2. The*

*development of in situ upper air measurements. Weather 64:302\u2013308. https://doi.org/10.1002/wea.484*

**Authors: -** Thank you for your rigorous review. The phrase in L46-47 of the preprint manuscript "Since the 19th century, the balloon-borne radiosonde has served as the primary tool for direct measurements of upper-air meteorological elements below 30 km and is extensively utilized globally (Gallice et al., 2011)"has been corrected to "The integration of radio technology with balloon-borne soundings emerged in the 1930s, with early operational systems deployed in the 1940s; this technology has since served as a primary tool for direct measurements of upper-air meteorological elements below 30 km and is widely utilized on a global scale (Pettifer 2009; Gallice et al., 2011)." on Page 2, L48-51 of the revised manuscript.

*5)48 "For over a century" - "For nearly a century"*

**Authors: -** Thank you for your rigorous review. The reason is same as "11)95 'more than a century' - 'about 80 years' ". The phrase in L48 of the preprint manuscript "For over a century" has been corrected to "For nearly a century"on Page 2, L52 of the revised manuscript.

*6)49 "one measurement" - "one profile"*

**Authors: -** Thank you for your rigorous review. The phrase in L49 of the preprint manuscript "one measurement" has been corrected to "one profile" on Page 2, L53 of the revised manuscript.

*7)54 "as Russia's decrease from twice-daily" - "as Russia's temporary decrease from twice-daily"See https://www.ecmwf.int/sites/default/files/elibrary/2016/18147-global-radiosonde-network-under-pressure.pdf*

**Authors: -** Thank you for your suggestion.The phrase in L54 of the preprint manuscript "such as Russia's decrease from twice-daily to once-daily launches in 2015" has been corrected to "such as Russia's temporary reduction of launches from twice to once daily in 2015" on Page 2, L58 of the revised manuscript.

*8)61-63 "The United States and other countries have successfully assimilated airborne sonde data into their model systems, resulting in a 20%-40% reduction in errors in hurricane trajectory predictions (Stephen et al., 2013; Wang et al., 2015)." "20%-40% reduction in errors" seems too high to me for modern NWP systems with good assimilation of satellite data, Majumdar (2016) already referenced is more balanced.*

**Authors: -** Thank you for your suggestions. Based on the findings of Majumdar's 2016 paper, the sentence in L61-63 of the preprint manuscript "The United States and other countries have successfully assimilated airborne sonde data into their model systems, resulting in a 20%-40% reduction in errors in hurricane trajectory predictions" has been revised to "The United States and other nations have successfully assimilated airborne radiosonde data into their model systems, leading to a 10–15% reduction in errors associated with hurricane trajectory predictions, particularly over oceanic regions where data is sparse."on Page 2, L67-70 of the revised manuscript.

9)68 *"The upper-air balloon system, known as ValBal, developed by the Stanford Space Program in the United States, has accomplished multi-day flights at altitudes ranging from 10 km to 25 km (Sushko et al., 2011)." I hadn't heard of ValBal, I suspect that it was a short-lived research system. On the contrary the WindBorne system (Johnson et al, 2024) is active and currently providing data on the GTS Johnson, A., Wang, X., Hutchinson, T., & Creus-Costa, J. (2024). Impact of WindBorne observation assimilation on prediction of a TPV merger case from THINICE. Journal of Geophysical Research: Atmospheres, 129, e2024JD041395. https://doi.org/10.1029/2024JD041395*

**Authors: -** As you can see, it has been over eight years since the ADDRS was initially developed in 2017. The early research on "Valbal" A. Sushko, A. Tedjarati, J. Creus-Costa, S. Maldonado, K. Marshland and M. Pavone, "Low cost, high endurance, altitude-controlled latex balloon for near-space research (ValBal)," 2017 IEEE Aerospace Conference, Big Sky, MT, USA, 2017, pp. 1-9, https://doi.org/10.1109/AERO.2017.7943912."Abstract:High-altitude balloons in near space offer the possibility of affordable scientific experimentation and hardware testing for outer space missions. In this paper we present a novel, low cost high-altitude balloon system that achieves multi-day flight using inexpensive latex balloons by automatically venting lifting gas and dispensing ballast to maintain altitude. Traditionally, superpressure balloons have been used for high-altitude scientific missions; however, despite their long endurance and payload capacity in the tens of kilograms, their cost is in excess of tens of thousands of dollars. Latex balloons are significantly less expensive, typically costing little more than a hundred dollars, but in normal use fly for only a couple hours, rising until reduced atmospheric pressure causes the balloon to stretch beyond its limits. Precision-weighted latex balloons have demonstrated multi-day flights, but such systems cannot change altitude while aloft and offer minimal payload capacity (measuring in tens of grams). Our system, known as ValBal, offers altitude control capabilities exceeding those of a superpressure balloon at a two order of magnitude reduction in cost. ValBal can

stabilize anywhere in its operational range of 10-25 km altitude, and can execute scheduled or remotely commanded altitude transitions during flight. In its current iteration, ValBal can be conFigd to accommodate payloads on the order of 10000 cubic centimeters and 2 kilograms. In June 2016, a ValBal demonstration mission flew for over 70 hours continuously, surpassing the previous world record of 57 hours, for the longest duration of a latex balloon flight. ValBal has flown twice more since then, including a flight of almost 80 hours. Planned developments will seek to improve the endurance to a week and increase the payload interface capabilities for scientific missions." which does indeed somewhat deviate from the current mainstream trend of international space exploration.

Therefore All content related to "Valbal" has been revised to "WindBorne". The sentence in L68-71 of the preprint manuscript "The upper-air balloon system, known as ValBal, developed by the Stanford Space Program in the United States, has accomplished multi-day flights at altitudes ranging from 10 km to 25 km (Sushko et al., 2011). The system maintains altitude by automatically venting gas and dispensing ballast, allowing the latex balloon to gradually rise while slowly drifting." has been amended to "The WindBorne emerged in 2019 from the cradle of the Stanford Space Program. Once launched, the WindBorne balloon regulates its altitude by jettisoning sand ballast and releasing gas, enabling multiple round-trip vertical sounding flights from below 20 km down to near the surface, with an average flight duration of seven days.(Johnson, 2024)"on Page 2,3, L75-78 of the revised manuscript.

10)81 "descent data" see also Ingleby et al (2022) Ingleby, B., Motl, M., Marlton, G., Edwards, D., Sommer, M., von Rohden, C., Vömel, H., and Jauhiainen, H.: On the quality of RS41 radiosonde descent data, Atmos. Meas. Tech., 15, 165-183, https://doi.org/10.5194/amt-15-165-2022, 2022.

**Authors: -** Thank you for your suggestions. The reference for this literature has been supplemented on Page 3, L90 of the revised manuscript.

11)95 'more than a century' - 'about 80 years' (prior to the 1940s there were tiny numbers of research flights that mostly relied on someone finding and returning the instrument package and   recorded data)

**Authors: -** Thank you for your suggestions. (Haig et al., 1958) Page 2 "It was not until 1927 that the first balloon-borne device transmitted signals back from the stratosphere. Such devices were named 'radiosondes' to distinguish them from 'balloon-sondes' (French for 'sounding balloon'). By 1940 radiosondes had almost completely replaced the aircraft meteorographs for the daily soundings used for weather forecasting. Pilot-balloons tracked by theodolite still provided the only means

of obtaining wind data above the surface—and still do in many parts of the world." So we also think 'more than a century' and 'about 80 years' may be not precise enough, Therefore,we have refer to your advice"5)preprint 48 "over a century" modify to "nearly a century" on Page 3, L102 of the revised manuscript.

*12)106 "ascends at approximately 400 meters per minute" convert to metres per second*

**Authors: -** Thank you for your suggestions. According to the operational of CMA upper-air meteorological observation specification, "meters per minute" is conventionally used as the unit. "Dual-mode balloon" techniques in line with WMO (World Meteorological Organization): Guide to Meteorological Instruments and Methods of Observation, Volume III - Observing Systems, WMO-No. 8, Page 464, pp., https://library.wmo.int/idurl/4/41650, 2025. "Usually, a rate of ascent between 5 and 7 m/s' is desirable in order to minimize the time required for observation; it may also benecessary in order to provide sufficient ventilation for the radiosonde sensors." So modify it to "at a rate of ascent between 5 and 7 m/s" on Page 4, L113 of the revised manuscript.

*13)186 "The ascent phase of meteorological sounding generally lasts less than one hour." It is usually 1.5 to two hours for most radiosondes. Or is this a statement about RDSS?*

**Authors: -** Thank you for your suggestions. We originally thought that the burst height of the commonly used 750g weather balloon or the outer balloon of the "Dual-mode balloon" (650g) as inflation carrier for the CMA upper-air meteorological observation, at an ascent rate of 5-7m/s, is about one hour. However, we overlooked the CMA's seven GUAN station also need use the 1600g weather balloon, which is the burst height of 36km and can generally last for 1.5-2 hours. Thank you for your meticulous feedback. The sentence "The ascent phase of meteorological sounding generally lasts less than one hour." has been revised to 'The ascent phase of meteorological sounding typically lasts between 1.5 and 2 hours."on Page 4, L191 of the revised manuscript.

*14)193 "hydrogen loss"  It should have been stated earlier that hydrogen is the lifting gas.*

**Authors: -** We agree with the reviewer, In the WMO (World Meteorological Organization): Guide to Meteorological Instruments and Methods of Observation, Volume III - Observing Systems, WMO-No. 8, Page 466, pp.,

https://library.wmo.int/idurl/4/41650, 2025. "The two gases most suitable for meteorological balloons are helium and hydrogen." It is necessary to point out that hydrogen is gase for inflation. So, "and filled with hydrogen" has been added on Page 4, L112 of the revised manuscript.

15)232 *"The positioning accuracy of SoC is 0.8 meters horizontally and 1.3 meters vertically" 1.3 meters vertically sounds better than I would expect - please provide some evidence for this. Is the accuracy worse near the surface (due to reflection or ducting of the signals)?*

**Authors: -** Thanks for the suggestion. We provide evidence that the static vertical positioning accuracy is 1.3 meters, while the dynamic vertical positioning accuracy is 2.8 meters. The positioning accuracy affected worse near the surface (due to reflection or ducting of the signals). And the u-blox-G7020 is 2.5 meters, but the GNSS chip is a old type for radiosondes. It's not fair put this result in a published research article. So we omit it(preprint manuscript L232"The positioning accuracy of SoC is 0.8 meters horizontally and 1.3 meters vertically").

[Figure]

*16)242-243 "The GTH3 participated in WMO UAII2022(Upper-Air Instrument Intercomparison Campaign organized by the World Meteorological Organization (WMO) and co-organized by the Deutscher Wetterdienst (DWD) in 2022)" Their table lists Tianjin Huayuntianyi Special Meteorological Sounding Tech. Co., Ltd. HT-GTS(U)2-1 (GTH3). I think it is worth giving the manufacturer name here and probably the alternate radiosonde name (HT-GTS(U)2-1). Is the current "GTH3" the same as that tested in UAII2022 or a development of it?*

**Authors: -** Thanks for the suggestion. It has been revised to 'The HT-GTS(U)2-1 (GTH3) of Tianjin Huayuntianyi Special Meteorological Sounding Tech. Co., Ltd. participated in' on Page 8, L236-245. The current CMA operational radiosonde "GTH3" is the same type as that tested in UAII2022.

*17)Table 4 caption "The planetry boundary layer (PBL) ranges from 2 to 7 kilometers" - should say "Free Troposphere" other parts of the caption also need changing.*

**Authors: -** We agree with the reviewer. It should in alignment with IOM-143 Report of UAII2022. It has been modified to "The planetary boundary layer (PBL) ranges from the surface to 2 kilometers; and Free Troposphere (FT) ranges from the 2 kilometers to7 kilometers on Page 9,10, L250-251.

*18)"4.1 Field experiment ..." What is the distribution (above ground level) of the lowest descent point received? This depends on the orography and how close to a receiving station it lands.*

**Authors: -** We thank the reviewer for this careful reading. From March to September 2021,the Field experiment in the middle and lower reaches of the Yangtze River Region is total 1,772 achieved successful drifting number of samples, which is the distribution (above ground level) of the descent section height lower than 1km was 465, accounting for 26.2% of the 1772 groups of samples with successful drifting. In the early tests, the receivers were spaced far apart and were affected by various factors such as terrain obstruction. We predict the trajectory in Fig. R3 Spatial distribution of the landing points (colored dots) for sounding stations (black triangles) across China from February 2022 to January 2024. Different colors represent different months. The relevant results are actively optimizing the layout of the receiver. It is expected that the data of the subsequent descent section will be less than 1km by more than 50%.

| Station | WMO Station No | Mean initial descent altitude (m) | Mean final descent altitude (m) | the descent section height lower than 1km |
|---|---|---|---|---|
| Yichang | 57461 | 24326.27 | 5609.74 | 86 |
| Wuhan | 57494 | 25498.95 | 6437.54 | 111 |
| Changsha | 57687 | 26365.63 | 5022.52 | 49 |

| | | | | |
|---|---|---|---|---|
| Ganzhou | 57993 | 24701.05 | 5022.52 | 38 |
| Anqing | 58424 | 25862.66 | 5884.57 | 105 |
| Nanchang | 58606 | 26044.70 | 6312.61 | 76 |

19) 359 "the fifth generation of ECMWF (ERA5) was used to evaluate the QC data of RDSS (Minola et al. 2020; Roja et al. 2011)," "the fifth generation of ECMWF reanalysis (ERA5, Hersbach et al, 2020) was used to evaluate the quality of RDSS data," I'm not sure how relevant Minola et al is. Roja et al predates ERA5 and doesn't appear min the references. Hersbach H et al (2020) The ERA5 global reanalysis. Q J R Meteorol Soc 146:1999-2049. https://doi.org/10.1002/qj.3803

**Authors: -** As suggested by the reviewer, we have been removed two less relevant references, and the authoritative reference to Hersbach's ERA5 has been re-cited on Page 24, L646-652 of revised manuscript.

20) Table 5. 'EAR5' should be 'ERA5'; 'troposphere top' should presumably be 'tropopause', units of RH are "%RH". '{SD of] radiosonde data and ERA5' should be 'radiosonde data minus ERA5'

**Authors: -**We thank the reviewer for this careful reading. We have corrected the typo on Page 15, L375 of revised manuscript and redraw the Table 5.

21) 411-412 "With the four-dimensional ensemble forecast error is introduced into the CMA global data assimilation system, and the H-4DEnVar assimilation scheme is developed." The English here is strange/unclear. What is the H in H-4DEnVar?

**Authors: -** We thank the reviewer for this careful reading. H is meaning "hybrid". We have corrected the description , "To avoid the tangent linear and adjoint models, the four-dimensional ensemble forecast error is introduced into the CMA global data assimilation system, and the Hybrid-4DEnVar assimilation scheme is developed" on Page 17, L426-427 of revised manuscript.

23) Fig 10 caption. Stage that positive values indicate benefit when descent data is used. Precipitation thresholds in mm?

**Authors: -** We thank the reviewer for this careful reading. We have corrected the Figure 10 on Page 18, L426-427 of revised manuscript.

[Figure]

[Figure]

**Figure 10. Improvement rates of cumulative precipitation thresholds (mm) predictions for 0-12 hours (a) and 12-24 hours (b) in the Down test compared to the control test.**

*24)449-452 "Targeted observations ... " This is overstated. They have "sometimes been a frontier field" The impact of targeted observations is almost always less than one or two years work on improving the baseline NWP system (Lorenc pers comm) - and improving the baseline tends to reduce the benefit of targeted observations. "Majumdar & Sharanya, 2016" should be "Majumdar, 2016" Two quotes for his 'Conclusions and Discussion': "The fact that many results are inconclusive is not surprising, for several reasons. ..." "In summary, the initial vision that the Global Observing System would ultimately be supplemented by networks on an adaptive basis, or even optimized into a fully adaptive network, has not been realized."*

**Authors: -** Thanks for the suggestion. According to statistics, China is directly affected by the landfall of 7 to 8 typhoons on average each year, and another 4 to 6 coastal typhoons cause wind and rain impacts. To continuously monitor the new characteristics of typhoon path northward movement and intensity increase, conducting target observations such as ADDRS is becoming increasingly important for CMA 's operational sounding. The sentence 'Targeted observation have sometimes been one of frontier field in atmospheric science research.' has been revised in L463-464 of revised manuscript., and a summary from the conclusions of Majumdar (2016) has been added: 'Additionally, Majumdar (2016) highlights that advancements in numerical weather prediction (NWP) systems-such as improved data assimilation techniques and enhanced model resolution—have reduced the marginal contributions of individual observing systems. Furthermore, the evaluation of targeted observations is constrained by factors including flow-dependent conditions, limited sample sizes, and inconsistencies in verification metrics. Therefore, cost-effective strategies for targeted observations necessitate exploration through multi-agency coordinated observing system experiments, such as FSO studies.'

*25)500 '5.3.3 Targeted observations experiment of Typhoon'*

*I was surprised by this section because surely the biggest impact on TC forecasting comes from improving the analysis and forecast one, two or three days before landfall. In the online discussion a response was "The RDSS can cover the South China Sea area" where would they be launched from?*

**Authors: -**Thank you for your suggestions. In the online discussion a response is mostly based on Wen, Q. S., Zhang, X. F., Hu, S., Zhao, P. T., Zhong, S. X., Liu, Z. Y., Zhao, Z. K., Liang, J. H., Dai, G. F., Zhang, C. Z., Li, M. J., Huang, L.: Collaborative assimilation experiment of Beidou navigation radiosonde and drone-dropped radiosonde based on CMA-TRAMS, Atmos. Oceanic Sci. Lett., 18(2), https://doi.org/10.1016/j.aosl.2024.100555, 2025. Based on the trajectory prediction system and the wind speed and direction of the typhoon, Implement Targeted observations experiment of Typhoon"Haikui". Such as the typhoon passes through the northern area of the South China Sea area, target observations have been launched from Guangdong Province's four stations of ADDRS. Yangjiang, Heyuan, Qingyuan and Shantou. Among them, Qingyuan and Shantou(GTS exchange stations).

*26)For another view see*

*Magnusson, L., Majumdar, S.J., Dahoui, M.L., Bormann, N., Bonavita, M., Browne, P.A., et al. (2025) The role of observations in ECMWF tropical cyclone initialisation and forecasting. Quarterly Journal of the Royal Meteorological Society, 151(768), e4924. Available from: https://doi.org/10.1002/qj.4924*

*"The strongest impact came from withholding of microwave radiance observations ..." "Scatterometer data improved the forecasts ...""While the aircraft-borne data (dropsonde, flight-level data) were mostly beneficial, further investigations on how to best use this data in the TC core are needed."*

**Authors: -** Yes, we are carry out consistently use dropsonde and so on extensive mobile observations exploring the TC core(Zhang., et al., 2021; Liu, L. H et al., 2022; Wen., et al., 2024 ). This is a very meaningful research direction. Thanks for the suggestion. (Magnusson.,et al., 2025) has been added in L480 of revised manuscript.

*28)552 'century-old' - '20th century'*

**Authors: -**Thanks for the suggestion. As suggested by the reviewer, we have modified "century-old" to "20th century" on Page 22, L567 of revised manuscript.

*29)556 'RDSS is essentially a mature system.' I would say it is close to maturity (but not there yet). No data have been provided on the GTS yet as far as I know. Some of my questions in the general comments need answering/clarification.*

**Authors: -** Thanks for the suggestion. The ADDRS still have some technical issues that need to be resolved. Engineering technology requires continuous practice and application to realize its value in a long-term, dynamic iterative process. 'RDSS is essentially a mature system.' is omit on Page 22, L570 of revised manuscript.

*30)References*

*Where the papers are in Chinese please indicate this.*

*Many readers will not be able to read such papers, I suggest removing some of the less relevant ones.*

**Authors: -** Thanks for the suggestion. The Chinese literature has been marked as "(in Chinese)" in the REFERENCES.

*31)632 "Majumdar, S. J.: A review of targeted measurements" this is the whole of the tile, delete ": targeted measurements to improve numerical forecasts of high-impact weather events over the past two decades, particularly during the THORPEX era (2005-14), are evaluated"*

**Authors: -** Thank you for your suggestion. It has been modified to "Majumdar, S. J.: A review of targeted observations, Bull. Am. Meteorol. Soc., 97, 2287-2303, https://doi.org/10.1175/BAMS-D-14-00259.1, 2016." on Page 26, L687-688 of revised manuscript.

*32)660 "Tan et al ...   doi:10.3969/j.issn.1004-4965.2006.01.003." I couldn't find this despite the doi.*

**Authors: -** Thank you for your suggestion. Your rigorous academic review has greatly benefited us. It has been modify to "Tan, X. W., Chen, D. H., Zhang, Q. H.: An impact study of a new type of data of adaptive or targeting observation on typhoon forecast, J. Trop. Meteorol., 22(1), 18–25, https://doi.org/10.3969/j.issn.1004-4965.2006.01.003, 2006." on Page 26, L710-712 of revised manuscript.